# CUS3D: A New Comprehensive Urban-Scale Semantic Segmentation 3D Benchmark Dataset

## Abstract

With the continuous advancement of smart city construction, the availability of large-scale and semantically enriched datasets is essential for enhancing the machine's ability to understand urban scene. When dealing with large-scale scene, mesh data has a distinct advantage over point cloud data, as it can provide inherent geometric topology information and consume low memory space. However, existing publicly available large-scale scene mesh datasets have limitations in scale and semantic richness, and cannot cover a wider range of urban semantic information. In addition, existing large-scale 3D datasets lack various types of officially provided annotation data, which limits the widespread applicability of benchmark applications and can easily result in label errors during data conversion. To address these issues, we propose a comprehensive urban-scale semantic segmentation benchmark dataset. This dataset provides finely annotated point cloud and mesh data types for 3D, as well as high-resolution original 2D images with detailed 2D semantic annotations. It is well suited for various research pursuits on semantic segmentation methodologies. The dataset covers a vast area of approximately 2.85 square kilometers, containing 10 semantic labels that span both urban and rural scenes. Each 3D point or triangular mesh in the dataset is meticulously labeled with one of ten semantic categories. We evaluate the performance of this novel benchmark dataset using 8 widely adopted deep learning baselines. The dataset will be publicly available upon the publish of the paper.

## 1 Introduction

In the 3D world, large-scale scene semantic segmentation assigns specific meanings to every object within a scene, thereby aggregating the complex and diverse environmental information into several specific semantic categories to facilitate planning and research. Large-scale scene semantic segmentation helps machines accurately identify object features in large scenes and plays a crucial role in advancing the development of fields such as scene-level robot navigation (Valada et al., 2017), autonomous driving (Geiger et al., 2013), urban planning (Chen, 2011), spatial analysis (Yaagoubi et al., 2015), and urban fluid simulation (García-Sánchez et al., 2014).

To improve the machine's ability to recognize the semantics behind segmented urban data, researchers often need ample real data from which the machine can learn. In large-scale semantic segmentation methods, deep-learning-driven algorithms are currently mainstream. With the maturity of convolutional neural networks (CNNs), deep networks that can directly process point clouds (e.g., PointNet (Qi et al., 2017a), PointNet++ (Qi et al., 2017b), KPConv (Thomas et al., 2019), and RandLA-Net (Hu et al., 2020)) have reached new heights in point-cloud semantic segmentation, especially for city-scale outdoor scenes.

Deep neural networks that directly process mesh data, such as MeshCNN (Hanocka et al., 2019) and MeshNet (Feng et al., 2019), only achieve good results in the semantic segmentation of small-scale objects; they perform poorly in mesh semantic segmentation in large-scale scenes. This phenomenon may be due to the existence of few publicly available large-scale scene mesh datasets with fine-grained semantic annotations. Currently, the only publicly available city-level scene mesh datasets are Hessigheim 3D (Kölle et al., 2021) and SUM (Gao et al., 2021). However, Hessigheim

3D (Kölle et al., 2021) has a scene area of only 0.13 square kilometers, and some areas are not annotated, while SUM (Gao et al., 2021) only contains one type of data (mesh) and has only six semantic categories. Thus, it lacks the richness of rural and suburban scene information, making it unsuitable for research on smart urban and rural development planning. Multi-view-based methods for semantic segmentation in large scenes (Kundu et al., 2020) are also among the mainstream approaches for 3D semantic segmentation, but no publicly available large-scale scene 3D semantic segmentation datasets with relevant multi-view 2D images currently exist, which somewhat restricts the development of multi-view-based algorithms. Therefore, the current large-scale scene semantic segmentation datasets face the following challenges: **First**, the scene semantics type is limited and does not include various scenes of cities, suburbs, and rural areas. **Second**, These datasets do not provide multiple official data types with precise labels, and errors or omissions in data labeling may occur during the process of sampling and conversion of 3D data such as point clouds and meshes.

Therefore, we propose a new example of urban-level outdoor scene semantic segmentation with rich semantic labels and diverse data types. Our dataset, called CUS3D, includes two types of 3D data: point clouds and meshes, which are annotated with both the real color information and semantic labels. We also provide the original high-resolution aerial images used for 3D reconstruction. Fine-grained semantic annotations are performed on both the 3D data and 2D images, creating a benchmark for current research on 2D and 3D semantic segmentation algorithms. The point-cloud data in this dataset consists of approximately 152,298,756 3D points, while the mesh data consists of 289,404,088 triangles. It covers an area of approximately 2.85 square kilometers and includes 10 semantic categories: building, road, grass, car, high vegetation, playground, water, building site, farmland, and ground. At the same time, we conduct benchmark tests on six classic semantic segmentation deep networks to ensure the suitability of the CUS3D dataset. Compared to existing 3D datasets, CUS3D has the following contributions:

1. CSU3D is a comprehensive urban-scale outdoor scene semantic segmentation benchmark that includes two types of 3D data, point clouds and meshes, both of which are assigned with detailed semantic labels. It is suitable as a benchmark for testing almost all of the mainstream 3D semantic segmentation algorithms.

2. CUS3D dataset provides high-resolution 2D aerial images captured by an unmanned aerial vehicle (UAV), and the images are meticulously annotated with semantic labels. The semantic labels of the 2D images are geometrically consistent with the 3D data. The 2D images in this dataset can be used for research in areas such as 3D rendering, 3D reconstruction, and remote sensing image semantic segmentation.

3. CUS3D dataset has richer semantics and covers the semantic information of almost all of the urban scenes. CUS3D also includes the semantic information of suburban and rural scenes, such as farmland and building sites. This benefit provides new opportunities for large-scale urban applications, such as smart cities and urban planning.

## 2 RELATED WORK

Owing to the diversity of acquisition methods and reconstruction techniques, the data types of 3D data are more varied than those of 2D data. Among them, point clouds, meshes, and voxels are common types of 3D data.

### 2.1 3D POINT CLOUD DATASET

Four categories of existing 3D point cloud datasets exist, based on their scale and scene type: 1) object-level 3D point cloud datasets, 2) indoor scene-level 3D point cloud datasets, 3) outdoor road-level 3D point cloud datasets, 4) large-scale urban-level 3D point cloud datasets.

**Object-level 3D point cloud datasets**: These datasets mainly consist of individual objects of different categories. These include ModelNet (Wu et al., 2015), ShapeNet (Chang et al., 2015), Object-Net3D (Xiang et al., 2016), PartNet (Mo et al., 2019), ShapePartNet (Yi et al., 2016), ScanObjectNN (Uy et al., 2019), and the newly released OmniObject3D (Wu et al., 2023). These object-level 3D datasets are commonly used for performance competitions of algorithms such as visual classification and segmentation.

**Indoor scene-level 3D point-cloud datasets**: These types of datasets are usually collected using depth scanners. These datasets include ScanNet (Dai et al., 2017), SUN RGB-D (Song et al., 2015), NYU3D (Silberman et al., 2012), SceneNN (Zhou et al., 2017), and S3DIS (Armeni et al., 2017), released by Stanford University. These datasets are widely used in the early development stage of deep networks for directly processing point clouds. They are suitable for testing 3D semantic segmentation algorithms for small scenes.

**Outdoor road-level 3D point-cloud datasets**: These datasets are usually collected through LiDAR scanners and RGB cameras, including ParisLille-3D (Roynard et al., 2018), SemanticKITTI (Behley et al., 2019), SemanticPOSS (Pan et al., 2020), A2D2 (Geyer et al., 2020), Waymo Open Dataset (Sun et al., 2020), Toronto-3D (Tan et al., 2020), nuScenes (Caesar et al., 2020), CSPC-Dataset (Tong et al., 2020), Lyft Dataset (Lyft, 2019), Oakland3D (Munoz et al., 2009), Paris-rue-Madame (Serna et al., 2014), iQmulus (Vallet et al., 2015), and KITTI (Geiger et al., 2012). Additionally, some datasets simulate road scenes through synthesis to obtain more accurate semantic labels, such as the Synthia Dataset (Ros et al., 2016). These datasets are often applied in research on autonomous driving, including target recognition and semantic segmentation.

**Large-scale urban-level 3D point-cloud datasets**: Large-scale urban-level 3D point-cloud datasets are typically collected in two ways. The first way is through aerial LiDAR scanning, including DALES (Varney et al., 2020), LASDU (Ye et al., 2020), DublinCity (Zolanvari et al., 2019), Semantic3D (McCormac et al., 2017), and ISPRS (Rottensteiner et al., 2012). The second way is to obtain 2D images through UAV oblique photography and then obtain point-cloud data through 3D reconstruction technology, including Campus3D (Li et al., 2020) and SensetUrban (Hu et al., 2021). This type of dataset is now more often acquired using the second way. Compared to point-cloud data obtained from aerial LiDAR scanning, this technology offers color information, which can better reflect the structure of urban scenes and is beneficial to smart city planning.

## 2.2 3D MESH DATASET

Mesh is a common representation of 3D data. Currently, relatively few publicly available 3D datasets are based on mesh data. We classify them into two categories: 1) object-level 3D mesh datasets and 2) scene-level 3D mesh datasets.

**Object-level 3D mesh datasets**: This type of dataset is typically composed of various instances of single objects. In this type of dataset, individual parts of a single object are annotated and segmented, and multiple classes of objects in different poses are categorized. Examples of such datasets include Human Body (Maron et al., 2017), COSEG (Wang, 2023), SHREC11 (Lian et al., 2011), MSB (Shape Analysis Group, 2005), and mesh-type 3D data from ModelNet (Wu et al., 2015). The end-to-end semantic segmentation deep networks for 3D Mesh data are still in the early stage of development. Existing methods such as MeshCNN (Hanocka et al., 2019) and MeshNet (Feng et al., 2019) perform semantic segmentation and classification on mesh data by defining different feature descriptors. The open-source methods validate the reliability of the algorithms using object-level mesh data.

**Scene-level 3D mesh datasets**: Scene-level 3D mesh datasets are typically obtained through high-quality 3D reconstruction techniques and have geometric topology and high-resolution real scene textures. Currently available indoor 3D mesh datasets include ScanNet (Dai et al., 2017), Matterport 3D (Chang et al., 2017), Replica Dataset (Straub et al., 2019), and 3D-FUTURE (Fu et al., 2021). However, relatively few large-scale mesh datasets for outdoor urban-level scenes exist. ETHZ CVL RueMonge (Riemenschneider et al., 2014) is the first benchmark dataset provided in mesh format that is related to urban scenes. However, owing to errors in multi-view optimization and fuzzy boundaries, many incorrect labels exist in this dataset. Hessigheim 3D (Kölle et al., 2021) is a small-scale urban scene semantic segmentation dataset that covers an area of only 0.19 square kilometers. The semantic labels of the mesh data are transferred from the point-cloud data labels, so approximately 40% of the area is unmarked. Some non-manifold vertices also exist, making it difficult to directly apply this dataset. SUM (Gao et al., 2021) is the latest publicly available outdoor large-scale urban-level 3D mesh dataset known. The dataset covers an area of about 4 square kilometers and includes six semantic categories of urban scenes. However, the SUM (Gao et al., 2021) dataset only contains mesh data, which are relatively limited in terms of data type and semantic

categories. The dataset does not include semantic categories of suburban village scenes for urban and rural planning, so it has certain limitations in practical applications.

In summary, most existing publicly available large-scale 3D datasets consist of a single data type and have limited semantic richness. Especially for large-scale mesh datasets, the number of such datasets is small, and some datasets have partial semantic omissions or marking errors owing to semantic annotation strategies, which affect their use. With the continuous development of smart city planning and other fields, a comprehensive 3D semantic segmentation dataset with multiple data types and rich urban scene semantic information has become meaningful. In the following sections, we introduce a new urban-level outdoor large-scale semantic segmentation comprehensive dataset with diverse data types and rich, accurate semantic annotation.

Table 1: Comparison of existing 3D urban benchmark datasets

| Name | Year | Data type | Spatial size | Classes | Points/Triangles | RGB | Platforms |
|---|---|---|---|---|---|---|---|
| ISPRS (Rottensteiner et al., 2012) | 2012 | Point cloud | - | 9 | 1.2 M | No | ALS |
| DublinCity (Zolanvari et al., 2019) | 2019 | Point cloud | $2.0\ km^2$ | 13 | 260 M | No | ALS |
| DALES (Varney et al., 2020) | 2020 | Point cloud | $10.0\ km^2$ | 8 | 505.3 M | No | ALS |
| LASDU (Ye et al., 2020) | 2020 | Point cloud | $1.02\ km^2$ | 5 | 3.12 M | No | ALS |
| Campus3D (Li et al., 2020) | 2020 | Point cloud | $1.58\ km^2$ | 24 | 937.1 M | Yes | UAV camera |
| SensetUrban (Hu et al., 2021) | 2021 | Point cloud | $6.0\ km^2$ | 13 | 2847.1 M | Yes | UAV camera |
| Oakland3D (Munoz et al., 2009) | 2009 | Point cloud | $1.5\ km$ | 5 | 1.6 M | No | MLS |
| Paris-rue-Madame (Serna et al., 2014) | 2014 | Point cloud | $0.16\ km$ | 17 | 20 M | No | MLS |
| iQmulus (Vallet et al., 2015) | 2015 | Point cloud | $10.0\ km$ | 8 | 300 M | No | MLS |
| Semantic3D (McCormac et al., 2017) | 2017 | Point cloud | - | 8 | 4000 M | No | TLS |
| Paris-Lille-3D (Roynard et al., 2018) | 2018 | Point cloud | $1.94\ km$ | 9 | 143 M | No | MLS |
| SemanticKITTI (Behley et al., 2019) | 2019 | Point cloud | $39.2\ km$ | 25 | 4549 M | No | MLS |
| Toronto-3D (Tan et al., 2020) | 2020 | Point cloud | $1.0\ km$ | 8 | 78.3 M | Yes | MLS |
| ETHZ RueMonge (Riemenschneider et al., 2014) | 2014 | Mesh | $0.7\ km$ | 9 | 1.8 M | Yes | Auto-mobile camera |
| Hessigheim 3D (Kölle et al., 2021) | 2021 | Point cloud&Mesh | $0.19\ km^2$ | 11 | 125.7 M/36.76 M | Yes | Lidar&camrea |
| SUM (Gao et al., 2021) | 2021 | Mesh | $4\ km^2$ | 6 | 19 M | Yes | UAV camera |
| CUS3D (Ours) | 2023 | Point cloud&Mesh | $2.85\ km^2$ | 10 | 152.3 M/289.4 M | Yes | UAV camera |

## 3 CUS3D DATASET

### 3.1 UAV CAPTURES AERIAL IMAGES

In the past, 3D data (such as point clouds) were typically obtained by using LiDAR scanners in cars. With the development of UAV tilt photography and 3D reconstruction technology, UAV has gained significant advantages in mapping. To obtain high-resolution aerial image sequences, we select the DJI M300 RTK and equipped it with the advanced Penta-camera SHARE PSDK 102S V3 for 2D aerial image acquisition. To ensure comprehensive coverage of the designated measurement area, the UAV follows a preplanned flight path and captures images at specified intervals of 2 seconds. The flight altitude of the UAV is 100 meters, and the weather conditions are good without any obstructive factors such as clouds, haze, or strong winds, providing the UAV with good visibility and stable flight conditions. The flight control system is automated. Owing to the limited capacity of the UAV's onboard batteries, every battery group can support approximately 30 minutes of flight time. The capture of images for the entire area is composed of multiple separate flight missions executed in parallel.

To validate the accuracy and quality of the evaluation data, we use a high-precision real-time kinematic (RTK) GNSS for geo-referencing the captured aerial images. In addition, six ground control points are manually set up in the real scene to validate the accuracy of the subsequent 3D reconstruction data. The positions of these six checkpoints are the locations where UAV takes off for data collection. Please refer to the appendix for the specific location diagram.

### 3.2 3D MODEL RECONSTRUCTION

In the reconstruction process of mesh data, the first step is to perform aerial triangulation on the collected 2D image data to determine the position and pose of the camera and minimize the reprojection error. Then, the results of the aerial triangulation are used to construct a large-scale 3D scene using 3D reconstruction techniques. However, large lakes exist in our CUS3D dataset, where the 3D water bodies do not meet the Lambertian assumption during reconstruction. To address

this issue, we use semi-automated water body restoration techniques to obtain a complete 3D mesh model. The reconstructed CUS3D mesh dataset is divided into 93 blocks, consisting of a total of 289,404,088 triangle meshes and covering an area of approximately 2.85 square kilometers. The details of the reconstructed mesh and point-cloud scene are shown in Figure 1.

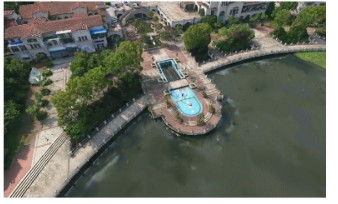 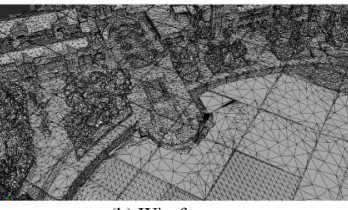 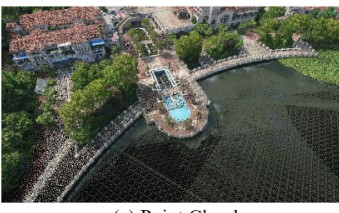

(a) Texture mesh         (b) Wireframe         (c) Point Cloud

Figure 1: 3D reconstruction data display: (a) 3D reconstructed textured mesh; (b) Texture-less mesh with white base model; (c) 3D point cloud reconstructed by sampling.

In the reconstruction process of point-cloud data, we down-sample the mesh data to obtain point-cloud data. We transfer the real color information to the vertices of the point-cloud data through a nearest neighbor search. The generated dense colored point cloud consists of 152,298,756 3D points. The point cloud density is approximately 0.15 meters, covering an area of about 2.85 square kilometers.

To verify the accuracy of the 3D reconstruction data, we place six control points manually on the ground while capturing the original images. After completing the construction of the 3D model, we report a median re-projection error of 0.82 pixels and a median reprojection error for every control point of 0.62 pixels, which falls within a reasonable error range. Therefore, we consider the constructed 3D model to be accurate and reliable.

## 3.3 3D SEMANTIC LABEL ANNOTATION

Our CUS3D dataset contains a variety of data types and diverse semantic information. It consists not only of semantic labels for urban scenes but also semantic labels for rural scenes, such as agricultural landscapes and suburban infrastructure. As modern cities grow and smart city initiatives expand, it has become increasingly important to incorporate urban-rural and suburban planning and construction into smart city development. By integrating the semantic information from rural scenes into urban-level 3D models, we can make accurate, informed decisions for smart urban-rural planning.

The CUS3D dataset has abandoned the previous fixed urban semantic labeling strategy and adopted a dynamic urban semantic labeling strategy, considering the objects' economy, functionality, and temporality for semantic differentiation, making it more applicable to practical applications such as urban planning, transportation planning, and architectural decision-making. According to the standard definition of semantic categories, we identify 10 meaningful semantic categories in the CUS3D dataset. Every 3D point or triangle mesh in the dataset is assigned a unique semantic label using annotation tools. Further, all of the annotated data have undergone two rounds of manual cross-checking. Figures 2 and 3 show an example of our dataset's semantic annotation, Figure 4 shows the errors found during manual cross-checking and promptly corrected, and Table 1 compares the Urban3D dataset with other existing 3D urban datasets. The specific 10 object classes in our benchmark dataset are as follows:

1. Building: including commercial buildings and residential homes;
2. Road: including streets, asphalt roads, nonmotorized lanes, and parking lots;
3. Car: including small cars and large buses;
4. Grass: including lawns and small shrubs;
5. High vegetation: including fir trees and banyan trees;
6. Playground: including basketball courts, athletic tracks, and amusement park;
7. Water: including lakes, and canals;
8. Farmland: including cultivated fields, undeveloped land, and livestock enclosures;
9. Building sites: including construction material yards, and construction sites;
10. Ground: including cement surfaces, asphalt surfaces, and bare soil surfaces.

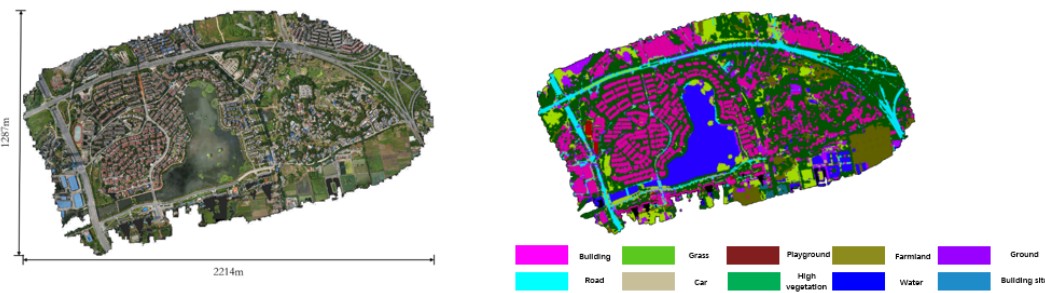

Figure 2: CUS3D mesh benchmark dataset semantic annotation schematic diagram.

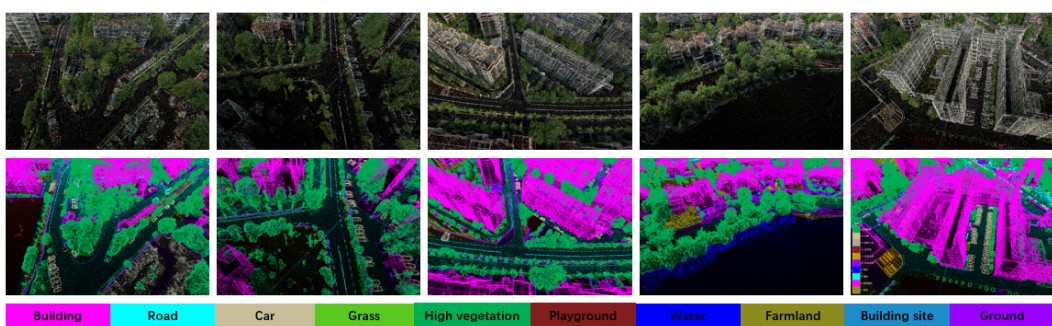

Figure 3: CUS3D point-cloud benchmark dataset example.

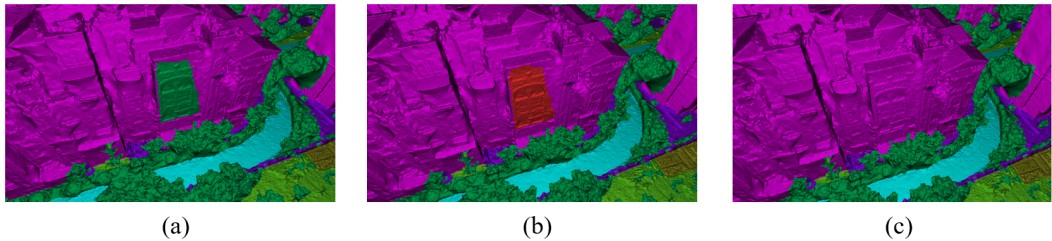

|(a)|(b)|(c)|

Figure 4: In the manual cross-check, the incorrectly labeled wall is found and adjusted. (a) Discover the wall that is incorrectly labeled. (b) Select the object with the lasso tool. (c) Assign the correct label to the incorrectly labeled wall.

## 3.4 2D IMAGE DATA ANNOTATION

In addition to providing 3D meshes and point-cloud data with rich semantic labels, CUS3D also annotates high-resolution raw aerial images captured by a UAV, giving them detailed semantic labels. There is no publicly available 3D semantic segmentation dataset with annotated 2D images. Providing both 2D and 3D data with semantic consistency is beneficial for the development of 3D rendering, 3D reconstruction, and semantic segmentation of remote sensing images.

We use an automated instance segmentation tool based on the SAM (Kirillov et al., 2023) model for intelligent semantic labeling. This technology is not limited to specific scenarios and can shorten the labeling cycle. We provide 4336 annotated 2D image data, divided into training, testing, and validation sets in an 8:1:1 ratio. The semantic labels of these images are the same as the semantic categories of the 3D models. They have geometric alignment in data annotation, and each image is annotated with high-quality pixel-level annotations and manual cross-checking, ensuring high annotation accuracy. Figure 5 shows the results of our 2D aerial image semantic annotation.

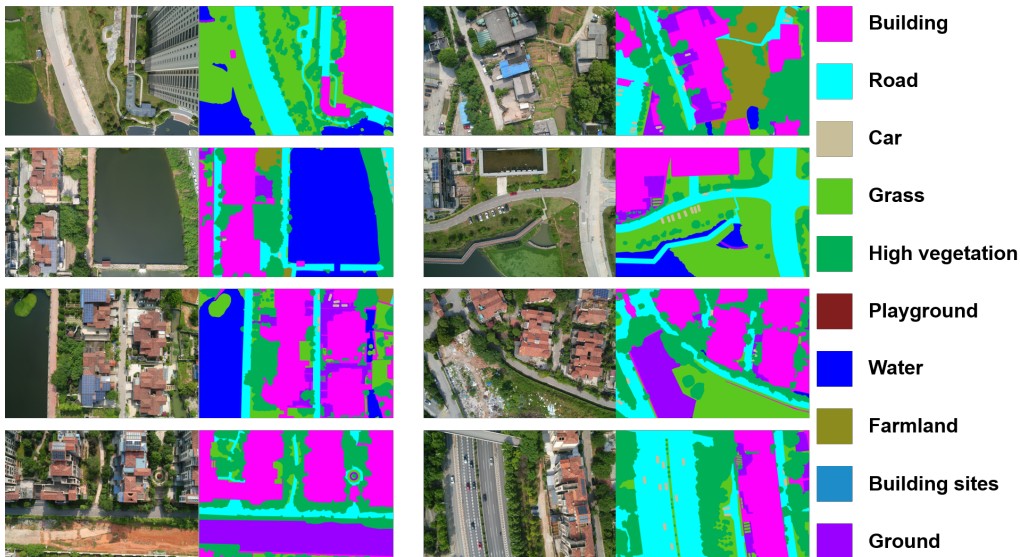

Figure 5: Partial high-resolution aerial images and their detailed semantic labels, including 10 semantic categories.

# 4 EXPERIMENT

## 4.1 DATA SPLIT

To use the dataset as a benchmark for semantic segmentation tasks, we divide all of the blocks from the CUS3D dataset in a ratio of 8:1:1. Sixty-six blocks are used as training data; eight blocks as testing data; and eight blocks as validation data (see Figure 6). Owing to the varying sizes of triangles in the 3D mesh data, when evaluating the semantic information of the mesh data, we calculate the surface area occupied by the triangles, rather than the number of triangles. For every of the 10 semantic classes, we calculate the total surface area of the corresponding class in the dataset to represent its distribution, as shown in Figure 6. From the class distribution chart, it can be observed that certain classes, such as the car and playground, account for less than 5% of the total surface area, while categories like building, lake, and high vegetation account for over 70% of the total surface area. Thus, imbalanced class distribution poses a significant challenge in supervised learning-based semantic segmentation.

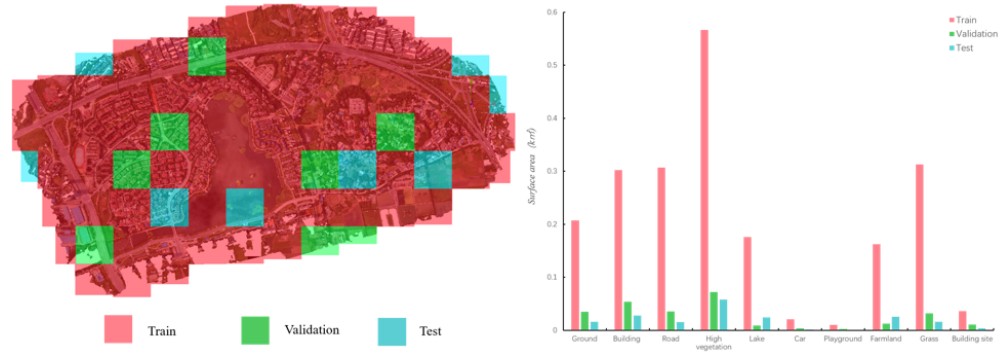

Figure 6: Left: The distribution of the training, testing, and validation dataset. Right: Semantic category distribution in both the training, validation, and testing dataset.

## 4.2 BASELINES

Currently, to our knowledge, no open-source deep learning framework exists that can directly handle large-scale texture mesh data in urban 3D spaces. However, in urban environments, point clouds and meshes have almost identical fixed attributes. Both types of data share the same set of feature vectors in the same scene. Therefore, we choose point-cloud data from the dataset as the input for testing deep neural networks. The test results show consistent accuracy for both types of 3D data. We carefully select seven representative methods as benchmark tests for our CUS3D dataset, including supervised and weakly supervised semantic segmentation methods. In addition, we selected the DeepLabV3+ network to perform semantic segmentation benchmark testing on 2D aerial imagery data.

1. PointNet (Qi et al., 2017a). This is a groundbreaking work for directly processing unordered point clouds.
2. PointNet++ (Qi et al., 2017b). This is an improved version of PointNet that includes the extraction of local features.
3. RandLA-Net (Hu et al., 2020). This work directly deals with the semantic segmentation of large-scale point clouds and ranked first in the Semantic3D dataset challenge.
4. KPConv (Thomas et al., 2019). This method references kernel-point convolution for semantic segmentation and achieves good performance on the DALES dataset.
5. SPGraph (Landrieu & Simonovsky, 2018). This is one of the methods that uses super point graphs to process large-scale point-cloud data.
6. SQN (Hu et al., 2022). This is one of the weakly supervised methods that applies a small number of labels for semantic segmentation of large-scale point clouds.
7. Stratified Transformer (Lai et al., 2022). This is a point cloud semantic segmentation method based on the powerful transformer module.
8. DeepLabV3+(Chen et al., 2018). This is the method for benchmark testing for semantic segmentation of 2D aerial images.

## 4.3 EVALUATION METRICS

In the selection of evaluation metrics for semantic segmentation, we choose accuracy, recall, F1 score, and intersection over union (IoU) for every category, similar to the metrics for existing benchmarks. For the overall region testing evaluation, the metrics we adopt are overall accuracy (OA), mean accuracy (mAcc), and mean intersection over union (mIoU).

## 4.4 BENCHMARK RESULTS

To ensure fairness in the experiment, we maintain the original experimental environment for every baseline and conduct the experiments on an NVIDIA Tesla V100 GPU. The hyper-parameters of every experimental network are adjusted based on the validation data to achieve the best results possible. Table 2 lists the quantitative results of different network experiments after testing, and on our dataset benchmark, KPConv (Thomas et al., 2019) has the best overall performance in all of the overall performance indicators, with an mIoU of 59.72%, an OA of 89.42%, and an mAcc of 97.88%. The reason KPConv (Thomas et al., 2019) has good overall and individual category performance is that the network has a good ability to learn local features, and it can capture geometric and semantic information in the data. SQN (Hu et al., 2022) has good overall performance, especially in the few categories with fewer points (e.g., Car, Playground, Building site), thanks to the use of shell-based representations to process point data, which offers good rotation and scaling invariance. At the same time, it can be seen from the results that our dataset does not perform well on SPGraph (Landrieu & Simonovsky, 2018), mainly because it cannot capture local feature information of aggregated data well in large scenes, which also limits the wide application of the CUS3D dataset.

Our CUS3D dataset provides additional color information to the network compared to the large-scale 3D datasets collected by LiDAR, which is helpful for better understanding the real world. However, this may cause the network to overfit. To explore the impact of RGB information on the final semantic segmentation results, we select five baseline networks for comparative experiments. In every group of baseline comparative experiments, only coordinate information is input, or both the coordinate information and RGB information are input for training. Table 3 shows the quantitative experimental results of different feature inputs on five baselines. We can see that when coordinate

Table 2: Benchmark results of the baselines on CUS3D.Overall Accuracy (OA,%), mean class Accuracy (mAcc,%), mean IoU (mIoU,%), and per-class IoU (%) are reported. (Bold indicator only represents the highest value under the 3D baseline, the 2D baseline is not included for comparison.)

| | Ground | Building | Road | High vegetation | Lake | Car | Playground | Farmland | Grass | Building site | mIou | OA | mAcc |
|---|---|---|---|---|---|---|---|---|---|---|---|---|---|
| PointNet | **78.36** | 65.81 | **78.34** | 73.32 | 25.22 | 20.35 | 8.76 | 8.02 | 39.11 | 16.66 | 42.54 | 74.57 | 94.91 |
| PointNet++ | 25.78 | 80.36 | 57.77 | 86.18 | 57.52 | 36.13 | 11.38 | 40.88 | 40.17 | 13.28 | 55.16 | 84.52 | 96.90 |
| RandLA-Net | 15.29 | 76.18 | 49.52 | 78.42 | **62.30** | 49.89 | **35.91** | **42.11** | 39.74 | 29.86 | 52.98 | 77.12 | 95.42 |
| KPConv | 25.73 | **82.89** | 48.69 | **89.64** | 48.43 | 44.69 | 25.72 | 41.86 | 44.84 | 15.50 | **59.72** | **89.42** | **97.88** |
| SPGraph | 12.05 | 32.10 | 30.56 | 63.40 | 31.71 | 4.4 | 7.65 | 36.66 | 31.36 | 28.84 | 30.44 | 67.23 | 40.73 |
| SQN | 30.65 | 78.63 | 60.22 | 79.08 | 59.21 | 55.46 | 34.77 | 39.88 | **45.54** | 30.73 | 56.82 | 78.19 | 95.64 |
| Stratified Transformer | 60.23 | 70.02 | 75.44 | 83.36 | 58.47 | **58.66** | 32.45 | 39.66 | 34.59 | **39.33** | 55.22 | 86.45 | 96.67 |
| DeepLabV3+ | 81.21 | 80.25 | 82.47 | 92.55 | 65.12 | 59.56 | 38.68 | 46.74 | 45.18 | 40.02 | 63.18 | 90.02 | 98.04 |

information and color information are input at the same time, PointNet (Qi et al., 2017a), KPConv (Thomas et al., 2019), and RandLA-Net (Hu et al., 2020) achieve better performance. On the contrary, when only coordinate information is used as input, many urban categories cannot be recognized for segmentation, such as playground, building site, and car categories, which achieve poor segmentation performance. The segmentation performance of SPGraph (Landrieu & Simonovsky, 2018) mainly depends on geometric partitioning, so whether RGB information is available has little effect on its segmentation performance.

Table 3: Quantitative results of the five selected baselines on the CUS3D dataset.Overall Accuracy (OA,%), mean class Accuracy (mAcc,%), mean IoU (mIoU,%), and per-class IoU (%) are reported.

| | OA (%) | mAcc (%) | mIou (%) | Ground | Building | Road | High vegetation | Lake | Car | Playground | Farmland | Grass | Building site |
|---|---|---|---|---|---|---|---|---|---|---|---|---|---|
| PointNet(w/o RGB) | 54.17 | 90.83 | 20.43 | 38.67 | 23.47 | 59.20 | 22.58 | 12.17 | 0.72 | 0 | 3.15 | 21.85 | 2.18 |
| PointNet(w/ RGB) | 74.57 | 94.91 | 42.54 | **78.36** | 65.81 | **78.34** | 73.32 | 25.22 | 20.35 | 8.76 | 8.02 | 39.11 | 16.66 |
| PointNet++(w/o RGB) | 70.03 | 94.0 | 29.62 | 11.35 | 50.64 | 33.73 | 65.32 | 15.49 | 15.13 | 0 | 16.78 | 21.29 | 0.14 |
| PointNet++(w/ RGB) | 84.52 | 96.90 | 55.16 | 25.78 | 80.36 | 57.77 | 86.18 | 57.52 | 36.13 | 11.38 | 40.88 | 40.17 | 13.28 |
| RandLA-Net(w/o RGB) | 70.19 | 94.04 | 40.66 | 10.15 | 73.76 | 30.69 | 70.98 | 44.39 | 42.87 | 8.06 | 36.44 | 24.52 | 18.32 |
| RandLA-Net(w/ RGB) | 77.12 | 95.42 | 52.98 | 15.29 | 76.18 | 49.52 | 78.42 | **62.30** | 49.89 | **35.91** | **42.11** | 39.74 | 29.86 |
| KPConv(w/o RGB) | 85.05 | 96.27 | 44.52 | 17.46 | 77.12 | 37.55 | 84.41 | 25.68 | 34.07 | 12.56 | 24.53 | 30.43 | 0 |
| KPConv(w/ RGB) | **89.42** | **97.88** | **59.72** | 25.73 | **82.89** | 48.69 | **89.64** | 48.43 | 44.69 | 25.72 | 41.86 | 44.84 | 15.50 |
| SQN(w/o RGB) | 69.31 | 93.86 | 41.25 | 24.47 | 70.50 | 48.93 | 67.12 | 44.24 | 40.05 | 3.76 | 28.47 | 26.57 | 15.32 |
| SQN(w/ RGB) | 78.19 | 95.64 | 56.82 | 30.65 | 78.63 | 60.22 | 79.08 | 59.21 | **55.46** | 34.77 | 39.88 | **45.54** | **30.73** |
| SPGraph(w/o RGB) | 67.23 | 39.40 | 30.12 | 12.54 | 30.49 | 28.61 | 63.62 | 30.85 | 4.29 | 7.04 | 37.46 | 30.72 | 29.17 |
| SPGraph(w/ RGB) | 67.23 | 40.73 | 30.44 | 12.05 | 32.10 | 30.56 | 63.40 | 31.71 | 4.4 | 7.65 | 36.66 | 31.36 | 28.84 |

The results of the above benchmark test experiment show that our CUS3D dataset can be used as a benchmark for existing 3D semantic segmentation networks, which can help machines better understand 3D scene information. In future work, we will also conduct more baseline tests of traditional semantic segmentation methods to further improve our dataset.

## 5 CONCLUSION

This paper has introduced an urban-level outdoor large-scale scene dataset with diverse data types and rich semantic labels. The dataset includes two types of 3D data: point-cloud and mesh, covering an accurately annotated area of 2.85 square kilometers. It consists of 10 semantic labels, encompassing the semantic information of both the urban and rural scenes, serving as a benchmark for 3D semantic segmentation. The dataset provides raw 2D images captured by UAVs, accompanied by detailed semantic labels. These images can be used for research in areas such as 3D reconstruction and high-resolution aerial image semantic segmentation. Through extensive benchmark tests, we have verified the applicability of the CUS3D dataset to different semantic segmentation baselines and conducted experiments and discussions on whether RGB information can improve semantic segmentation performance. In future work, we will continue to expand the scene scale and semantic information of CUS3D. We hope that the CUS3D dataset can promote research progress in emerging cyber-physical fields, such as modern smart city construction and digital twins.

### ACKNOWLEDGMENTS

This work has been partially supported by the research project of the National Natural Science Foundation of China on cross-category semi-supervised image semantic segmentation methods.

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

# APPENDIX

## A    DETAILS OF THE DATA COLLECTION

Our dataset has been obtained by reconstructing 2D aerial image sequences using SFM technology, which recovers the camera extrinsics for every image. We have collected the 2D aerial image sequences through UAV oblique photography, using the DJI M300 RTK quadcopter and the five-lens oblique camera SHARE PSDK 102s. The resolution of each image is 6144×4096. Figure 7 shows the image acquisition equipment, and Table 4 describes the relevant parameters of the SHARE PSDK 102s camera.

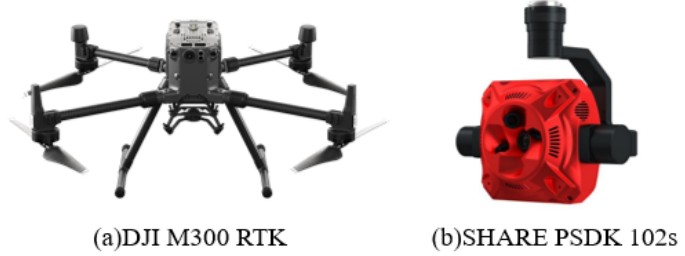

(a)DJI M300 RTK            (b)SHARE PSDK 102s

Figure 7: Left: DJI M300 UAV used for data collection; Right: SHARE PSDK 102s five-camera used for data collection.

Table 4: The parameters of the SHARE PSDK 102s camera

| Performance parameters | Numerical value |
| --- | --- |
| Lens number | 5 |
| Tilt angle | 45° |
| Image resolution | 6144×4096 |
| Focal lens | Downward:25 mm, sideward:35 mm |
| Sensor size | APS-C Format(23.5×15.6 mm) |

During the UAV shooting process, we use a preplanned Z-shaped route. The image capture interval is 2 seconds. The flight altitude for capturing the original aerial image sequence with UAV is 100 meters; the weather conditions on the day of data collection are clear, without any obstructive factors such as clouds, haze, or strong winds, providing the drone with a good field of vision, visibility, and stable flight conditions. The UAV is equipped with a five-lens camera, which can capture five images in different directions every time. One lens captures images from a downward angle, while the other four lenses capture images from the sides. To ensure the accuracy of the subsequent 3D reconstruction model, we set up six manual control points at six take-off points for verification during the shooting process. Figure 8 shows the locations of 6 artificial control points.

We conduct aerial triangulation, 3D reconstruction, and other technical steps on the 2D aerial image sequence to generate a 3D textured mesh model. To obtain the correspondence between 2D pixels and 3D points, we constructed an affine transformation. First, we computed the relative pose transformation between cameras by matching feature points in the images, and constructed the extrinsic matrix to establish the relationship between the 3D vertex coordinate system and the camera coordinate system. Then, using camera parameters such as focal length, we constructed the intrinsic matrix to establish the relationship between the camera coordinate system and the pixel coordinate system. Through coordinate transformation, we established the relationship between 2D pixels and 3D points. To identify the outliers, during the feature matching stage, the matching error of feature points between two images is eliminated using the RANSAC algorithm to remove outliers, avoiding impact on the pose estimation process. During the dense matching stage, the point cloud data is denoised and smoothed to remove outliers and sparse point clouds. Figure 9 shows the dimensions and overall appearance of the reconstructed 3D textured mesh model.

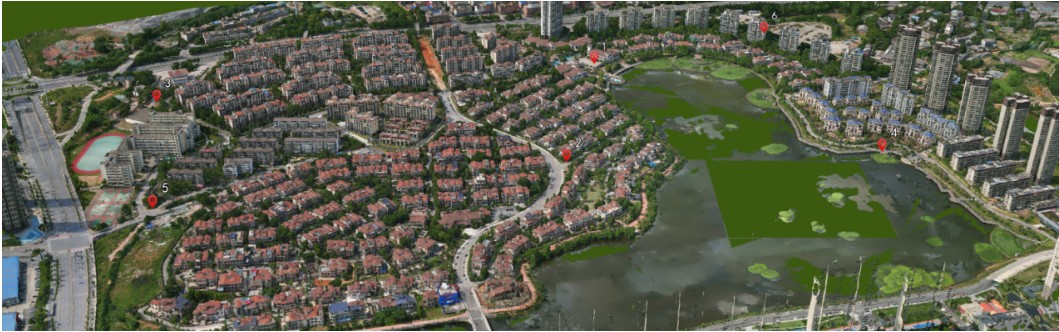

Figure 8: When capturing raw aerial images with a UAV, the positions of ground control points are manually set by the personnel on the ground.

The entire area covers an approximate land area of 2.85 square kilometers and consists of approximately 289 million triangular meshes. The terrain in this area is relatively flat, typical of suburban urban scenes, with a diverse range of features including buildings, roads, water systems, farmland, and vegetation. Note that the entire reconstructed scene is divided into 93 tiles, but only 89 tiles contain scene data. There are four tiles without any scene data, and their distribution is shown in Figure 10. These four blank tiles will be ignored in subsequent labeling and experiments. To provide a greater variety of 3D structures, we have also released point cloud data with true scene color information. The textured mesh data have been down-sampled to a point-cloud density of 0.15 meters, resulting in approximately 152 million points.

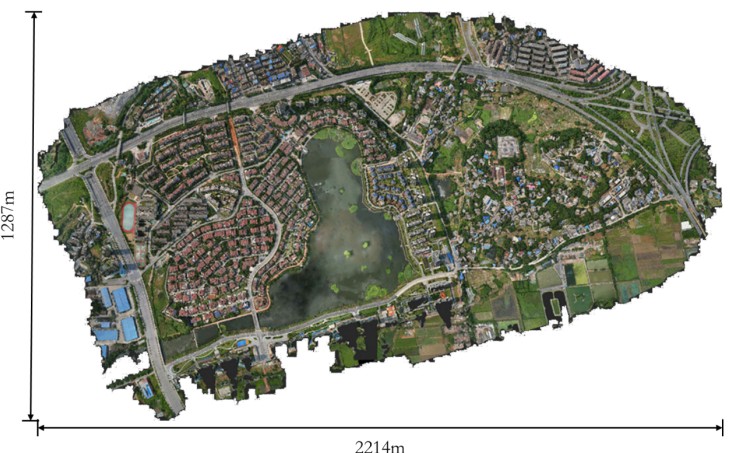

Figure 9: Original unmarked 3D texture mesh model. High-resolution texture mesh model reconstructed in 3D from a sequence of aerial images taken by a UAV, covering an area of 2.85 square kilometers.

## B    DETAILS OF THE DATA ANNOTATION

We use the DP-Modeler software for semi-automated 3D annotation to ensure that every triangle mesh is assigned the corresponding semantic labels. No triangle meshes are left unmarked. To prevent annotation errors and omissions, we ensure that all of the labels undergo two rounds of manual cross-checking to ensure accuracy. Figure 11 shows the issues discovered during the manual cross-checking process and promptly corrected. The semantic annotation process starts by inputting the source data, configuring the annotation categories according to requirements, and then performing manual labeling. After labeling has been completed, the classified data are subjected to quality inspection. Abnormal data are reclassified, resulting in labeled output data. The entire annotation

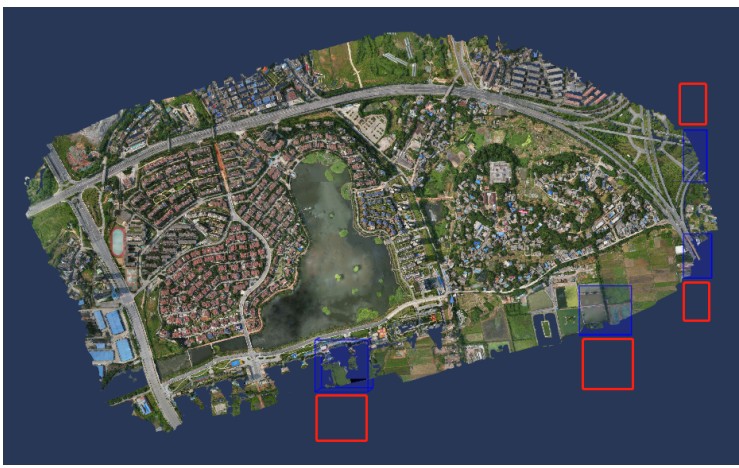

Figure 10: Blank Tile Distribution Chart (as indicated by the red box in the image).

process is illustrated in Figure 12, and the tool interface used in the semantic annotation process is shown in Figure 13.

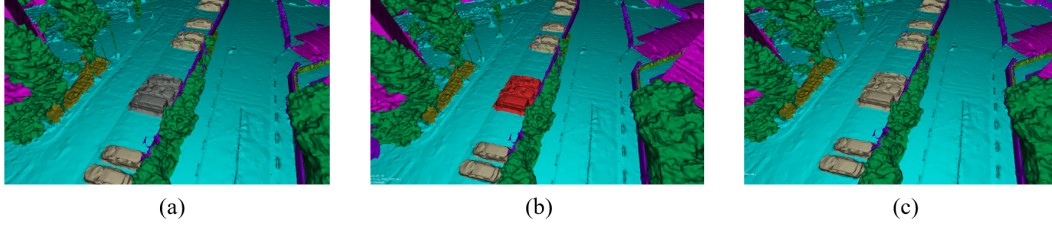

|  |  |  |
|:---:|:---:|:---:|
| (a) | (b) | (c) |

Figure 11: Artificial cross-checking detection of unmarked vehicles. (a) Unmarked vehicles detected; (b) Select unmarked vehicles; (c) Change the marking to vehicles.

In the selection of semantic labeling strategiesthe CUS3D dataset abandons the previous fixed urban semantic labeling strategy and adopts a dynamic urban semantic labeling strategy that considers both fully developed and developing semantic categories. For example, the categories of "road" and "ground" may have some overlapping characteristics. However, "road" belongs to fully developed functional objects, which can be used for urban transportation planning and peak traffic control optimization research. "Ground" belongs to undeveloped objects and labeling and recognition can help with early planning judgments in urban development. "Building sites" belong to the category of ongoing semantic objects and can be transformed into building in future semantic updates. The semantic labeling strategy of the CUS3D dataset considers the application, functionality, and temporal aspects of objects, making it more suitable for practical applications such as urban planning, transportation planning, and construction decision-making.

We classify different objects in the scene into 10 semantic categories. Considering that the scarcity of certain objects does not affect the planning and research of large-scale scenes, we categorize some high-granularity object information (e.g., pedestrians, utility poles, and solar panels) into their respective larger categories. These 10 semantic categories comprehensively represent the scene information in cities and suburbs. Every semantic label is assigned a specific color information. Table 5 provides the RGB values and grayscale values corresponding to every semantic label. Figure 14 shows the semantic labeling results of certain regions.

Regarding 2D image semantic labeling, the entire 2D image sequence consists of 10,840 images from 5 different perspectives. Due to the high similarity in the poses of four cameras tilted at 45°, we only selected 4,336 image sequences with a 90° top-down view and one 45° oblique view for semantic annotation. We have adopted the ABAVA data engineering platform developed by an outsourced company in our project team. The data labeling module of this platform provides an

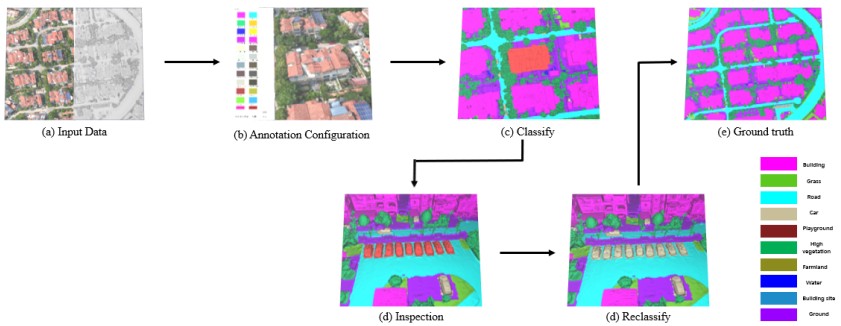

Figure 12: The pipeline of semantic tagging work.

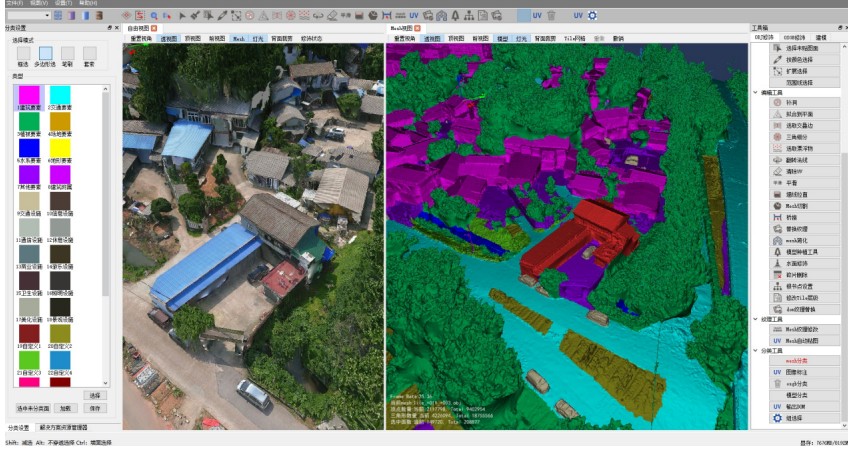

Figure 13: Semantic labeling software work interface.

automated instance segmentation tool based on the SAM algorithm. During the labeling process, the samples are first pre-labeled using the automated labeling module, and then manual adjustments are made to achieve high-precision pixel-level 2D image semantic labeling.

Table 5: Semantic label color information table

| Category | RGB value | Grayscale value |
|---|---|---|
| Building | (254,1,252) | 105 |
| Road | (0,255,255) | 179 |
| Car | (200,191,154) | 189 |
| Grass | (91,200,31) | 99 |
| High vegetation | (0,175,85) | 112 |
| Playground | (130,30,30) | 0 |
| Water | (0,0,255) | 29 |
| Farmland | (140,139,30) | 59 |
| Building sites | (30,140,201) | 201 |
| Ground | (154,0,255) | 75 |

## C  DETAILS OF THE EXPERIMENT

To verify the applicability of the CUS3D dataset on existing semantic segmentation networks, we conduct benchmark tests on seven 3D baseline methods. The hardware configurations are standard-

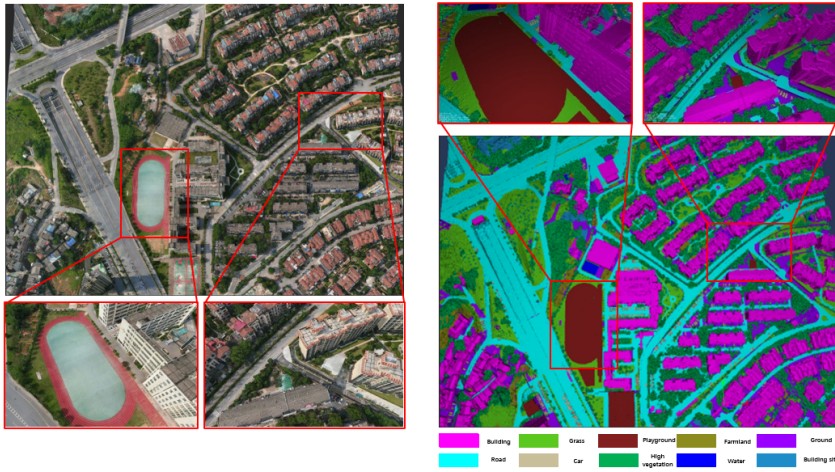

Figure 14: Partial region semantic labeling results. Left: Partial region original mesh; Right: Partial region semantic labeling results.

ized for the seven 3D benchmark tests, and the detailed hardware information is shown in Table 6. To ensure consistency in the benchmark tests, we ensure that the training, testing, and validation sets use the same regions and data quantities for the seven 3D test networks: PointNet(Qi et al., 2017a), PointNet++(Qi et al., 2017b), RandLA-Net(Hu et al., 2020), KPConv(Thomas et al., 2019), SPGraph(Landrieu & Simonovsky, 2018), SQN(Hu et al., 2022), and Stratified Transformer (Lai et al., 2022). Therefore, the dataset is uniformly partitioned. Additionally, owing to the influence of down-sampling on RandLA-Net(Hu et al., 2020) and SQN(Hu et al., 2022), some tile files have too few points. Therefore, the following seven LAS files are excluded when partitioning the dataset: Tile_+001+000, Tile_+001+006, Tile_+004+007, Tile_+005+000, Tile_+010+001, Tile_+011+007, and Tile_+012+002. In the main text, we show the distribution of the training, testing, and validation sets in the entire scene, as well as the distribution of every category in the mesh data. Table 7 shows the parameter settings of the baseline experiment.

Table 6: Baseline test experimental hardware environment configuration table.

| Name | Model |
|---|---|
| System | 4029GP-TRT2 |
| CPU | Intel Xeon 4210R |
| Memory | SAMSUNG 32GB DDR4 ECC 293 |
| System Disk | Intel S4510 |
| Data Disk | Intel S4510 |
| GPU | Nvidia Tesla V100 |

Table 7: Baseline experiment parameter settings

| | Epoch | Batch_size | Num_point | Learning_rate | Optimizer | Momentum | Parameters | Time |
|---|---|---|---|---|---|---|---|---|
| PointNet | 100 | 24 | 4096 | 0.001 | Adam | 0.9 | 0.97M | 6.4h |
| PointNet++ | 200 | 32 | 4096 | 0.001 | Adam | 0.9 | 1.17M | 6h |
| RandLA-Net | 100 | 16 | 4096 | 0.01 | Adam | 0.9 | 4.99M | 7.5h |
| KPConv | 500 | 10 | - | 0.01 | Adam | 0.9 | 14.08M | 8h |
| SPGraph | 500 | 2 | 4096 | 0.01 | Adam | 0.9 | 0.21M | 5.5h |
| SQN | 100 | 48 | 4096 | 0.01 | Adam | 0.9 | 3.45M | 7h |
| Stratified Transformer | 500 | 8 | 4096 | 0.001 | Adam | 0.9 | 34.63M | 41h |

During the data partitioning for the experiment, a total of 82 tiles were used for the training/validation/testing sets. Among them, 4 tiles did not have mesh data because they were located

at the edge of the measurement area, where the number of feature points for 3D reconstruction was too small to construct mesh data. Additionally, 7 tiles were not included in the dataset partition because, during the testing of the RandLA-Net network, the point cloud data of these 7 tiles had fewer than 4000 points, which did not meet the network's num_points requirement. In order to ensure the consistency of the input data for the network, these 7 tiles were deliberately ignored during the data partitioning.

To evaluate the performance of the CUS3D dataset, we choose ACC, Recall, F1 score, and IoU as evaluation metrics for every category. Overall, we choose mIoU, OA, and mAcc as evaluation metrics. In the main text, we present the metrics results for overall category testing and mIoU for every category. Tables 8, 9, 10, 11, 12, 13, and 14 show the test results for every semantic category in the seven networks.

Table 8: Evaluation metrics results of PointNet for every category

|  | Ground | Building | Road | High vegetation | Water | Car | Playground | Farmland | Grass | Building site |
|---|---|---|---|---|---|---|---|---|---|---|
| Acc(%) | 95.96 | 96.47 | 90.44 | 98.03 | 92.14 | 98.77 | 99.90 | 99.90 | 86.72 | 99.18 |
| Recall(%) | 88.35 | 80.95 | 91.77 | 88.80 | 40.42 | 25.32 | 9.98 | 9.98 | 50.53 | 20.10 |
| F1 Score(%) | 87.85 | 79.34 | 87.84 | 84.56 | 40.17 | 33.58 | 16.17 | 16.17 | 56.21 | 28.40 |
| IoU(%) | 78.36 | 65.81 | 78.34 | 73.32 | 25.22 | 20.35 | 8.76 | 8.76 | 39.11 | 16.66 |

Table 9: Evaluation metrics results of PointNet++ for every category

|  | Ground | Building | Road | High vegetation | Water | Car | Playground | Farmland | Grass | Building site |
|---|---|---|---|---|---|---|---|---|---|---|
| Acc(%) | 90.43 | 93.62 | 96.11 | 93.66 | 99.16 | 99.13 | 99.86 | 97.56 | 95.16 | 97.57 |
| Recall(%) | 42.37 | 89.62 | 83.14 | 91.72 | 65.01 | 46.83 | 18.64 | 72.42 | 53.42 | 16.09 |
| F1 Score(%) | 40.77 | 89.07 | 73.07 | 92.55 | 72.45 | 52.61 | 18.67 | 57.50 | 56.89 | 22.58 |
| IoU(%) | 25.78 | 80.36 | 57.77 | 86.18 | 57.52 | 36.13 | 11.38 | 40.88 | 40.17 | 13.28 |

Table 10: Evaluation metrics results of RandLA-Net for every category

|  | Ground | Building | Road | High vegetation | Water | Car | Playground | Farmland | Grass | Building site |
|---|---|---|---|---|---|---|---|---|---|---|
| Acc(%) | 88.24 | 94.49 | 95.12 | 90.15 | 98.67 | 97.64 | 97.32 | 96.67 | 93.31 | 96.59 |
| Recall(%) | 42.56 | 80.52 | 83.02 | 80.44 | 81.33 | 55.31 | 57.31 | 49.56 | 70.68 | 65.00 |
| F1 Score(%) | 26.04 | 86.38 | 65.17 | 87.88 | 76.43 | 66.04 | 51.87 | 58.84 | 56.42 | 45.38 |
| IoU(%) | 15.29 | 76.18 | 49.51 | 78.42 | 62.30 | 49.89 | 35.91 | 42.11 | 39.74 | 9.86 |

Table 11: Evaluation metrics results of KPConv for every category

|  | Ground | Building | Road | High vegetation | Water | Car | Playground | Farmland | Grass | Building site |
|---|---|---|---|---|---|---|---|---|---|---|
| Acc(%) | 93.79 | 95.00 | 97.50 | 93.94 | 99.51 | 99.26 | 99.90 | 98.21 | 96.18 | 98.76 |
| Recall(%) | 35.43 | 92.38 | 80.80 | 95.71 | 60.41 | 58.87 | 13.84 | 62.39 | 54.83 | 17.83 |
| F1 Score(%) | 40.19 | 90.54 | 65.16 | 94.52 | 63.81 | 60.27 | 12.58 | 57.87 | 61.53 | 25.14 |
| IoU(%) | 25.73 | 82.89 | 48.69 | 89.64 | 48.43 | 44.69 | 25.72 | 41.86 | 44.84 | 15.50 |

Table 12: Evaluation metrics results of SPGraph for every category

|  | Ground | Building | Road | High vegetation | Water | Car | Playground | Farmland | Grass | Building site |
|---|---|---|---|---|---|---|---|---|---|---|
| Acc(%) | 92.73 | 69.59 | 96.03 | 21.90 | 99.03 | 98.98 | 99.89 | 97.16 | 92.39 | 98.09 |
| Recall(%) | 14.24 | 38.48 | 42.68 | 90.88 | 40.51 | 4.75 | 8.91 | 51.30 | 42.14 | 42.56 |
| F1 Score(%) | 21.30 | 48.48 | 46.59 | 77.49 | 47.91 | 8.78 | 16.04 | 53.30 | 47.54 | 46.18 |
| IoU(%) | 12.05 | 32.10 | 30.56 | 63.40 | 31.71 | 4.40 | 7.65 | 36.66 | 31.36 | 28.84 |

Table 13: Evaluation metrics results of SQN for every category

|  | Ground | Building | Road | High vegetation | Water | Car | Playground | Farmland | Grass | Building site |
|---|---|---|---|---|---|---|---|---|---|---|
| Acc(%) | 86.09 | 95.36 | 95.06 | 91.98 | 98.28 | 98.04 | 97.87 | 96.52 | 93.22 | 96.45 |
| Recall(%) | 40.26 | 90.12 | 72.07 | 95.92 | 71.00 | 62.74 | 47.47 | 47.47 | 64.61 | 65.87 |
| F1 Score(%) | 46.44 | 87.76 | 74.74 | 87.73 | 73.81 | 70.70 | 50.94 | 56.37 | 61.96 | 46.32 |
| IoU(%) | 30.65 | 78.63 | 60.22 | 79.07 | 59.21 | 55.46 | 34.77 | 39.88 | 45.54 | 30.73 |

Table 14: Evaluation metrics results of Stratified Transformer for every category

|  | Ground | Building | Road | High vegetation | Water | Car | Playground | Farmland | Grass | Building site |
|---|---|---|---|---|---|---|---|---|---|---|
| Acc(%) | 91.22 | 94.43 | 96.12 | 94.67 | 99.34 | 99.21 | 99.85 | 97.89 | 96.12 | 97.88 |
| Recall(%) | 33.44 | 90.1 | 83.3 | 91.56 | 59.32 | 57.01 | 14.55 | 62.45 | 55.78 | 18.32 |
| F1 Score(%) | 41.34 | 90.22 | 66.19 | 90.44 | 62.63 | 52.78 | 12.89 | 57.85 | 57.03 | 22.68 |
| IoU(%) | 60.23 | 70.02 | 75.44 | 83.36 | 58.47 | 58.66 | 32.45 | 39.66 | 34.59 | 39.33 |

