# OpenReview forum: "CUS3D: A New Comprehensive Urban-Scale Semantic Segmentation 3D Benchmark Dataset"
_ICLR.cc/2024/Conference — ICLR 2024 Conference Withdrawn Submission_

### Official Review · Reviewer_a7Fe · 2023-10-13

**Soundness:** 3 good
**Presentation:** 3 good
**Contribution:** 3 good
**Rating:** 5
**Confidence:** 5

**Summary:**

This paper proposed CUS3D, which is a large-scale dataset for 3d semantic segmentation. It features various scenes and 2 data types, point clouds and mesh. 10 semantic categories are included in the dataet. 2d raw images are also annotated. 6 deep-learning based models are evaluated on the dataset to give baselines. Performance on models with or without RGB information as input is also evaluated.

**Strengths:**

The motivation of building a large dataset with diverse scenes and various data types is clearly stated.
The captured dataset is interesting, and implies a big collection effort.
The release of dataset is a nice contribution to the community.
6 baseline models are evaluated on the proposed dataset and overfitting is discussed.

**Weaknesses:**

CONBRITUTION: Since there already are many 3d datasets for semantic segmentation, and many of they are of drone images. The authors may want to state the contribution of their contribution more clearly. What's the weakness of existing datasets? Can the proposed dataset inspire research on some unexplored problems?

DATASET: The paper is overall sound and easy to follow, however, the authors may want to provide more detailed and clear descriptions for the proposed dataset.
1. Annotation accuracy: Is the dataset annotated by one annotator only? If not, please have everyone annotate some same areas and then measure the discrepancies between each person's annotations.
2. For drone images, scene depth is crucial. Please provide the drone’s flying height. Also, it would also be helpful if the author provide weather conditions while collecting the dataset, 3d reconstruction details(which algorithm or software is used? ). What's the resolution of CUS3d and existing datasets? A clear compare in table 1 may be more helpful.

EXPERIMENTS: The authors evaluate 6 baseline models on the dataset and discussed overfitting. From figure 6, I can see that the dataset is not evenly distributed on 10 categories. The authors may want to discuss influence of class imbalance on baseline models.

**Questions:**

1. The authors states that vast area and semantic richness as their main contribution. Does ‘semantic richness’ refer to variability in the content?
2. Is the train/val/test split randomly？

**Details Of Ethics Concerns:**

I don't have any ethics concerns of this paper.

---

> ### Author Response · Authors · 2023-11-22
> **Response to Reviewer a7Fe （1/5）**
>
> Thanks for the valuable comments. We provide explanations below to help reviewers better understand our work.
>
> A1：
>
> The contribution of the CUS3D dataset to the community is as a comprehensive semantic segmentation dataset with various data types, such as point clouds, meshes, and 2D images. It provides researchers with a benchmark dataset that contains multiple official data types, avoiding errors caused during the sampling and data conversion processes. Currently, no 3D semantic segmentation dataset exists that simultaneously includes point clouds, meshes, and 2D images. Existing outdoor scene mesh semantic segmentation datasets, such as the ETHZ CVL RueMonge dataset, produce many labeling errors owing to multi-view optimization and blurry boundaries. The SUM dataset only has six semantic classes, which is fewer compared to the CUS3D dataset. The Hessigheim3D dataset's mesh data labels were transferred from point-cloud data, resulting in approximately 40% of regions not being labeled. The semantic labels of both point-cloud and mesh data in the CUS3D dataset underwent manual cross-checking, ensuring no labeling errors or unlabeled areas. This dataset provides high-precision, diverse data types for the research community to use.

---

> ### Author Response · Authors · 2023-11-22
> **Response to Reviewer a7Fe （2/5）**
>
> Thanks for the valuable comments. We provide explanations below to help reviewers better understand our work.
>
> A2：
>
> The annotation strategy used in this dataset was multi-person joint quality control annotation. The entire dataset was annotated by three annotators, and a quality control process was set up. The three annotators simultaneously annotated three parts of the dataset. After each annotator had completed their annotation, they submitted it to the other two annotators for cross verification. After cross verification, a quality control expert conducted a comprehensive inspection and modified the corresponding error annotations as shown in Figure 11, achieving high-quality semantic annotation results.
>
> The flight altitude for capturing the original aerial image sequence with the drone was 100 m; the weather conditions on the day of data collection were clear, without any obstructive factors such as clouds, haze, or strong winds, providing the drone with a good field of vision, visibility, and stable flight conditions. For the 3D grid reconstruction, Mirauge3D software was first used to perform photogrammetric calculation on the images, and then the calculated results were input into the free version of DJI Terra software for triangulated mesh reconstruction. The mesh data spatial resolution of the CUS3D dataset was 3 cm, and the point-cloud data resolution is 53 points per m2. However, other 3D datasets have not fully disclosed their specific spatial resolutions, making it difficult to compare.

---

> ### Author Response · Authors · 2023-11-22
> **Response to Reviewer a7Fe （3/5）**
>
> Thanks for the valuable comments. We provide explanations below to help reviewers better understand our work.
>
> A3：
>
> The class imbalance issue in the CUS3D dataset was mainly reflected in the relatively low amount of data in the “car,” “playground,” and “building site” categories. This imbalance can make it difficult for some neural networks to converge, resulting in lower test metrics. Addressing this class imbalance is also one of the challenges in semantic segmentation for large-scale scenarios in the future.

---

> ### Author Response · Authors · 2023-11-22
> **Response to Reviewer a7Fe （4/5）**
>
> Thanks for the valuable comments. We provide explanations below to help reviewers better understand our work.
>
> A4：
>
> The semantic richness mentioned in this article mainly refers to the richness of semantic labels. The CUS3D dataset is a comprehensive outdoor scene dataset that includes both urban and rural scenes. In terms of label selection strategy, we chose to abandon the previous fixed urban semantic annotation strategy and instead adopted a city semantic annotation strategy with temporal evolution. This strategy considers both fully developed functional categories and undeveloped or under development categories. For example, “road” and “ground” may seem to have some overlap, but “road” belong to fully developed functional objects, which can be used for urban traffic planning, peak traffic flow control, and other urban management optimization research. “Ground” belongs to undeveloped objects, and annotating and identifying it can help with early-stage planning and judgment in urban construction. “Building sites” belong to objects that are under development, and appropriate “ground” can be planned and built so that when updating the map semantics in the future, they can be transformed into building sites and then into functional semantic categories such as buildings. At the same time, compared with existing outdoor scene semantic segmentation datasets, the CUS3D dataset adds “farmland” and other rural scene semantics, which is beneficial for research on smart city construction and urban-rural planning.

---

> ### Author Response · Authors · 2023-11-22
> **Response to Reviewer a7Fe （5/5）**
>
> Thanks for the valuable comments. We provide explanations below to help reviewers better understand our work.
>
> A5：
>
> The division of training/validation/testing was not random. The basis of the division strategy was to prioritize ensuring a balanced distribution of classes in the training and validation sets to avoid inaccurate model performance evaluation due to uneven data distribution. Based on this, the data were divided in a ratio of 8:1:1, and blocks of data that did not meet the experimental conditions were removed. The testing set consisted of the remaining tiles after excluding the training and validation sets, and the testing set was used to verify the model's generalization ability.

---

### Official Review · Reviewer_e913 · 2023-10-24

**Soundness:** 2 fair
**Presentation:** 3 good
**Contribution:** 3 good
**Rating:** 5
**Confidence:** 4

**Summary:**

This paper present a new dataset for semantic segmentation of wide-area urban 3D models.  The 3D model for the dataset was collected by flying a UAV over a 2.85 square kilometer area, collecting high resolution imagery, and applying a structure-from-motion method to build a surface mesh, including color texture.  The 3D model is then annotated manually, applying one of ten labels to every point in the mesh.  Source images were also independently annotated with a different set of class labels using a semi-automated approach leveraging segment anything (SAM model) to suggest segmentation.  The 3D model is divided into 93 tiles, and the tiles are divided to training, validation, and testing groups.  Experiments compare a number of recent method for 3D segmentation on this dataset using standard segmentation metrics and comparing the impact of using geometry and color versus geometry alone.

The main contribution of this paper is the large segmented 3D model, which will be useful in benchmarking future 3D segmentation work.

**Strengths:**

The key strength of this paper is the dataset itself and the potential value of this data in supporting quantitative evaluation of 3D segmentation methods in future publications.  The size and resolution of this model is quite large.  Constructing and annotating every surface of the mesh manually is an enormous undertaking.  Large, labeled 3D datasets are lacking in the community, so this one could be quite valuable.  The experiments are also useful and set baselines for future experimentation on this dataset.  Overall, the clarity and organization of the paper is good, with some exceptions noted below.

**Weaknesses:**

The primary weaknesses of this paper are the choice and definitions of the ten classes, the disconnect and inconsistency between 2D and 3D annotations and classes. Other weaknesses include claims about prior datasets being in a single format, lack of some important details on how the dataset was created, and some formatting issues.

One of the biggest issues with this paper are the inconsistency of the ten chosen labels and the overlap in what those classes cover.  The chosen labels appear to be a mix of functional classes (like building and road) and landcover classes (like grass and high vegetation).  Some classes seem too specific, like "Lake", which probably should be called "Water" and include various water bodies.  Some classes seem too broad, like "Ground", which seems to be a catch-all for everything not in one of the other classes.  Furthermore "Ground" seems to overlap with other labels creating ambiguity.  For example, "Road" contains asphalt roads and parking lots, but "Ground" also contains asphalt surfaces.  So How do annotators know how to apply these labels in a consistent way?  Similarly, "Ground" contains bare soil surfaces which are likely also found in "Building sites".  There seems to be a lot of overlap between "Grass" and "Farmland" definitions as well.

It's also quite strange that Section 3.4 presents 2D image annotation of the source imagery as an entirely independent labeling task.  It even uses a different set of 18 labels.  As far as I can tell, the 2D annotations are not used later an in any of the experiments.  So why are they included?  It seems like a significant oversight to use different labels for the 3D and 2D annotations and to assign these labels by independent processes.  If the classes were the same it would have been easy to generate the 3D labels from the 2D labels, or vice versus.  It would be very valuable to have a dataset with both 3D segmentation labels and 2D labels that are geometrically consistent with each other.  That is, if you project the labels from the 3D model into the image they are consistent with the 2D labels except in the case of moving objects like pedestrians and driving vehicles.  If Section 3.4 is not consistent with or related to the 3D model segmentation and not used in any further experiments then it is not relevant to the paper and should be removed.

In the introduction, the second claim about limitations of existing datasets is that these datasets are in a single format (point cloud or mesh).  This is a somewhat weak argument because any dataset that is provided as a mesh can be converted to a point cloud by sampling points on the surface.  This is exactly what is done in this paper.  I suppose the advantage of releasing the data as both a point cloud and mesh is that there is an official version of both the point cloud and the mesh for researcher to use in experiments.  So there is still value in releasing multiple formats, but it's not a big limitation of past mesh models.

There is some confusion about the number of tiles.  The paper says there are 93 tiles, but 4 are blank and Figure 10 shows the distribution of blank tiles on the periphery of the scene.  It's not clear why there are blank tiles, why there are only 4, and why this is important enough to have a figure showing where they are.  The paper then mentions an 8:1:1 split of training, testing, and validation.  However, it says there are 66 training, 8 test, and 8 validation blocks.  This is not exactly 8:1:1, which is fine, but it might be good to say "approximately 8:1:1".  More importantly, this only adds to 82, so what are the other 7 block use for?

At the bottom of page 8 it says "our dataset does not perform well on PointNet++ (Caesar et al. 2020)".  However, Table 2 shows that PointNet++ does perform well and SPGraph performs the worst.  Furthermore the citation of (Caesar et al. 2020) is not correct for neither PointNet++ nor SPGraph.

Other more minor issues are as follows:
- The paper mentions using SFM to construct the 3D model, but no detail are give about which SFM software/algorithm is used.
- Fonts in most figures and tables are too small
- It would be nice if the right of Figure 6 plotted train, test, and validation just like the left of the figure.
- Figure 7 has the wrong caption, a copy of the Figure 6.

**Questions:**

Please justify how the set of 10 ten class labels where selected and how you deal with ambiguities in the class definitions.  Why not reused the same classes used in prior work?

Please explain why there is a different set of classes for segmentation of the 2D images and why these are not consistent with the 3D segmentation classes.  Why is there also a different, independent process for annotating 2D images that doesn't seem to benefit the 3D annotations or vice versa?

What is the purpose of the blank tiles?  Why are they to begin with created?  Why are there only 4?  Why is their location important (Figure 10) if they are to be ignored?  What is the purpose of the extra 7 tiles that not blank but also not used in training, test, or validation?

**Details Of Ethics Concerns:**

This dataset was collected by flying a UAV over a large urban area.  It seems like this could be a privacy concern.  This may be mitigated by the resolution of the imagery being low enough that specific people and vehicles cannot be identified.  However, there was no statement in the paper about the potential for privacy concerns.

---

> ### Comment · Reviewer_e913 · 2023-11-21
> **Reviewers mostly in agreement**
>
> Three of the four reviewers for this paper had a common rating of "marginally below the acceptance threshold".  The authors did not provide any rebuttal to address any of the reviewers concerns, so I see no reason adjust my rating.  I continue to recommend that this paper needs further revision.

---

> ### Author Response · Authors · 2023-11-22
> **Response to Reviewer e913 （1/9）**
>
> Thanks for the valuable comments. We provide explanations below to help reviewers better understand our work.
>
> A1：
>
> The CUS3D dataset has abandoned the previous static urban semantic labeling strategy and adopted a new temporal functional urban semantic labeling strategy. This strategy considers both fully developed urban semantic categories and categories of undeveloped or currently developing objects.
>
> For example, although there may be some overlap between the categories “road” and “ground,” “road” belongs to fully developed functional objects, which can be used for urban traffic planning and peak traffic flow control. In contrast, “ground” belongs to undeveloped objects, and labeling and identification of this category can assist in the early planning and judgment of urban development. “Building site” belongs to the category of process-oriented semantic objects, and appropriate planning and construction on a suitable “ground” can later be updated to “building site” and transformed into the category of functional semantic objects such as buildings in future semantic updates of the map.
>
> Regarding the three semantic categories “grass,” “high vegetation,” and “farmland,” “farmland” belongs to the semantic of rural scenes and has special attributes related to human activities. The economic attributes in future urban expansion and activity planning are fundamentally different from the economic attributes of natural vegetation. The main difference between “high vegetation” and “grass” lies in their vertical height. The spatial costs and planning strategies required for urban greening construction are different, so they were separated into different categories.
>
> As for the definition of the “water” category, we believe that the term “water,” mentioned by the reviewing experts, represents a more extensive water body compared to “lake,” so we will modify this semantic category. In summary, the semantic labeling strategy of the CUS3D dataset considers the practical applications of objects, functionality, and temporal aspects, making it more suitable for urban planning, traffic planning, construction decision-making, and other practical applications.

---

> ### Author Response · Authors · 2023-11-22
> **Response to Reviewer e913 （2/9）**
>
> Thanks for the valuable comments. We provide explanations below to help reviewers better understand our work.
>
> A2：
>
> After initially collecting the aerial image sequence, we aimed to use these finely annotated 2D raw images for 3D reconstruction to obtain semantically meaningful 3D data. However, after the semantic annotation, the high similarity of pixel features would make feature point matching difficult during the triangulation process. Therefore, we decided to first reconstruct the original 3D data using traditional reconstruction methods and then perform semantic annotation. We also merged the initial 18 semantic categories into 10 categories that were easier to understand. Considering recommendations from expert reviews, we further aligned the semantic categories of the 2D images with the 10 semantic categories of the 3D data. Please refer to Figure 5 in the paper for the results of the semantic modification of the 2D image.  The semantic labels of the 2D images had high precision and were annotated at the pixel level. CUS3D provides 2D and 3D data that had geometrically consistent alignment on semantic labels while also making the originally captured aerial image sequence and pose parameters publicly available. To our knowledge, CUS3D is the first 3D dataset that publicly shares the entire original image sequence and pose parameters. These comprehensive data will play an important role in various aspects such as 2D aerial image semantic segmentation, 3D reconstruction, 3D rendering, and 3D semantic segmentation. Additionally, to demonstrate the usability of the 2D segmentation images, we conducted supplemental baseline tests using DeepLabV3+. The experimental results will be supplemented in the paper in the near future.

---

> ### Author Response · Authors · 2023-11-22
> **Response to Reviewer e913 （3/9）**
>
> Thanks for the valuable comments. We provide explanations below to help reviewers better understand our work.
>
> A3：
>
> The CUS3D dataset that we introduced is a comprehensive semantic segmentation dataset with multiple 3D data types, such as point clouds and meshes. Its purpose is to provide a benchmark dataset with various official data types to the research community, avoiding errors caused by researchers' own data conversion processes. For example, the semantic labels of the mesh data in the Hessigheim3D dataset were transferred from the point-cloud data labels, which resulted in approximately 40% of the areas being unmarked. In the CUS3D dataset, both the semantic labels of point-cloud and mesh data underwent manual cross-checking, ensuring no labeling errors or unmarked areas. Moreover, we also provided the original aerial image sequences and pose parameters of the images and performed 2D semantic annotation on some subsequences, providing high-precision, diverse data for the research community to use.

---

> ### Author Response · Authors · 2023-11-22
> **Response to Reviewer e913 （4/9）**
>
> Thanks for the valuable comments. We provide explanations below to help reviewers better understand our work.
>
> A4：
>
> The entire 3D scene of the CUS3D dataset consisted of 93 tiles, including four tiles without mesh data. These four tiles were not created first and were located at the edge of the measurement area. Owing to the insufficient number of feature points at the edge of the measurement area during 3D reconstruction, it was not possible to build a triangular mesh in that region. The specific positions of these tiles were not important. To ensure the original integrity of the provided data, we chose to include these four blank blocks and made an advanced explanation to avoid confusion for other researchers who might use this dataset in the future. However, these blocks themselves did not have any special significance.
>
> In the baseline experiment data partition, we intentionally ignored seven blocks. This is because testing of the RandLA-Net network showed that the point-cloud quantity of seven tile files was below 4,000, which did not meet the network's “num_points” setting. To ensure consistency of the input data for the six networks, we deliberately ignored these seven tiles during the dataset partition. However, we still choose to provide the complete dataset so that researchers in the community could perform specific data partitions according to their actual needs when conducting baseline experiments for new algorithms.

---

> ### Author Response · Authors · 2023-11-22
> **Response to Reviewer e913 （5/9）**
>
> Thanks for the valuable comments. We provide explanations below to help reviewers better understand our work.
>
> A5：
>
> This was our mistake; indeed, PointNet++ performed better than SPGraph in terms of experimental results. We have revised the incorrect content and references in the analysis of the experiments in the paper.

---

> ### Author Response · Authors · 2023-11-22
> **Response to Reviewer e913 （6/9）**
>
> Thanks for the valuable comments. We provide explanations below to help reviewers better understand our work.
>
> A6：
>
> First, Mirauge3D software was used for volumetric triangulation of images during grid 3D reconstruction. The results of the volumetric triangulation were then inputted into the DJI Mapping Free Edition software for triangle mesh reconstruction.
>
> Regarding the suggestions for modifying the charts, we have followed the suggestions and made corresponding revisions in the paper.

---

> ### Author Response · Authors · 2023-11-22
> **Response to Reviewer e913 （7/9）**
>
> Thanks for the valuable comments. We provide explanations below to help reviewers better understand our work.
>
> A7：
>
> We selected a city semantic labeling strategy that has temporal inference, rather than the previous fixed-city semantic labeling strategy. This strategy considers both fully developed functional and undeveloped or developing categories. For example, the categories of “roads” and “ground” may seem to have some overlap, but “roads” belong to fully developed functional objects, which can be used for urban transportation planning, peak traffic control, and other urban management optimization studies. “Ground” belongs to undeveloped objects, and labeling and identifying it can help with early planning judgments for urban development. Additionally, “building sites” belong to the category of process-based semantic objects under development, and suitable “ground” planning and construction can be carried out so that they can be transformed into building sites in future updates of the semantic map, eventually transitioning into functional semantic categories such as buildings. In summary, the semantic labeling strategy of the CUS3D dataset considers the applicability, functionality, and temporal inference of objects, making it more suitable for practical applications such as urban planning, transportation planning, and construction decision-making.

---

> ### Author Response · Authors · 2023-11-22
> **Response to Reviewer e913 （8/9）**
>
> Thanks for the valuable comments. We provide explanations below to help reviewers better understand our work.
>
> A8：
>
>  After initially collecting the aerial image sequence, we aimed to use these finely annotated 2D raw images for 3D reconstruction to obtain semantically meaningful 3D data. However, after the semantic annotation, the high similarity of pixel features would make feature point matching difficult during the triangulation process. Therefore, we decided to first reconstruct the original 3D data using traditional reconstruction methods and then perform semantic annotation. We also merged the initial 18 semantic categories into 10 categories that were easier to understand. Considering recommendations from expert reviews, we further aligned the semantic categories of the 2D images with the 10 semantic categories of the 3D data. Please refer to Figure 5 in the paper for the results of semantic modification of 2D images.  The semantic labels of the 2D images had high precision and were annotated at the pixel level. CUS3D provides 2D and 3D data that had geometrically consistent alignment on semantic labels while also making the originally captured aerial image sequence and pose parameters publicly available. To our knowledge, CUS3D is the first 3D dataset that publicly shares the entire original image sequence and pose parameters. These comprehensive data will play an important role in various aspects, such as 2D aerial image semantic segmentation, 3D reconstruction, 3D rendering, and 3D semantic segmentation.

---

> ### Author Response · Authors · 2023-11-22
> **Response to Reviewer e913 （9/9）**
>
> Thanks for the valuable comments. We provide explanations below to help reviewers better understand our work.
>
> A9：
>
> The entire 3D scene of the CUS3D dataset consisted of 93 tiles, including four tiles without mesh data. These four tiles were not created first and were located at the edge of the measurement area. Owing to the insufficient number of feature points at the edge of the measurement area during 3D reconstruction, it would not be possible to build a triangular mesh in that region. The specific positions of these tiles were not important. To ensure the original integrity of the provided data, we chose to include these four blank blocks and made an advanced explanation to avoid confusion for other researchers who might use this dataset in the future. However, these blocks themselves did not have any special significance.
>
> In the baseline experiment data partition, we intentionally ignored seven blocks. This is because testing of the RandLA-Net network showed that the point-cloud quantity of seven tile files was below 4,000, which did not meet the network's “num_points” setting. To ensure the consistency of the input data for the six networks, we deliberately ignored these seven tiles during the dataset partition. However, we still chose to provide the complete dataset so that researchers in the community could perform specific data partitions according to their actual needs when conducting baseline experiments for new algorithms.

---

### Official Review · Reviewer_ta5R · 2023-10-29

**Soundness:** 3 good
**Presentation:** 4 excellent
**Contribution:** 2 fair
**Rating:** 5
**Confidence:** 5

**Summary:**

The paper presents CUS3D, an urban-scale semantic segmentation 3D benchmark dataset intended to boost machine understanding of urban scenarios. Covering roughly 2.85 square kilometers, the dataset offers three data types including point clouds, images, and meshes with semantic annotations (i.e., 10 categories) across urban and rural scenes. It has been thoroughly tested with six point cloud semantic segmentation baselines, confirming its reliability for research.

**Strengths:**

The principal contribution of this paper lies in the provision of a novel dataset and benchmarks to the relevant community. The salient feature of this dataset is its offering of multiple annotated data formats, along with a substantial size that covers an area of nearly 3 $km^2$. Additionally, this paper presents a comprehensive review of existing 3D urban benchmark datasets, which hold a certain significance.

**Weaknesses:**

The reviewer appreciates the substantial effort made by the author in collecting, collating, and annotating data, thus providing meaningful resources for the community. However, the reviewer believes that while this work might be sufficient for a workshop paper, it would require additional contributions in terms of novelty and completeness to qualify as an academic article.

- Innovation: 1. The advantages of this dataset, in terms of timeliness and scale, are not strong when compared to existing datasets such as the earlier Campus3D and large-scale SansetUrban. 2. The method of constructing a photogrammetry 3D dataset is relatively common and has been explained in detail in Campus3D and SUM. And there are seldom technical contributions based on this dataset.

- Completeness: 1. The annotation of 2D areal images in this dataset employs cutting-edge methods SAM. However, there is a lack of detailed verification (accuracy and robustness of annotation) and other settings. 2. The paper proposes three annotated data formats but only showcases the baseline method based on point clouds. Considering the differences between 2D images and 3D, the baseline based on areal images should also be considered. 3. More technical details should be included. Please refer to question.

**Questions:**

1. The authors claim that the dataset has richer semantics and covers the semantic information of almost all of the urban scenes. As far as the reviewer knows, Campus3D and SensatUrban provide more categories than 10.

2. The reviewer indicates that the methods for baseline establishment should be updated including more SOTA methods like point cloud transformer (Lai, Xin, et al. "Stratified transformer for 3d point cloud segmentation." Proceedings of the IEEE/CVF Conference on Computer Vision and Pattern Recognition. 2022.)

3. As an important methodology, the SAM method for 2D annotation is not well-described, the author may clarify the parameters and settings.

4. The details for the point cloud segmentation baseline are not provided, including epoch, batch size, and other essential parameters. It is also important to specify how to do data preparation (e.g. sampling) for large-scale point clouds.

---

> ### Author Response · Authors · 2023-11-22
> **Response to Reviewer ta5R (1/6)**
>
> Thanks for the valuable comments. We provide explanations below to help reviewers better understand our work.
>
> A1:
>
> The number and semantic categories of point-cloud datasets like Campus3D and SensatUrban are large and diverse, respectively. However, the number of publicly available outdoor scene mesh semantic segmentation datasets is relatively small. As mentioned in Section 2, existing outdoor scene mesh semantic segmentation datasets such as ETHZ CVL RueMonge and Hessigheim have many labeling errors and unmarked regions owing to multi-view optimization and fuzzy boundaries. The SUM dataset only has six semantic categories, which is fewer compared with our CUS3D dataset. The contribution of the CUS3D dataset to the community is not to provide the point-cloud dataset with the most semantic labels, but as a comprehensive semantic segmentation dataset with multiple types of 3D data, such as point-cloud and mesh. It provides the community of researchers with a benchmark dataset with various official data types, avoiding errors arising out of the sampling and data conversion processes. For example, the semantic labels of the mesh data in the Hessigheim3D dataset were transferred from the point-cloud data labels, resulting in approximately 40% of the regions not being labeled. The semantic labels of both point-cloud and mesh data in the CUS3D dataset had also undergone manual cross-checking, ensuring no labeling errors or unmarked regions, thereby providing high-precision datasets for the community of researchers to use.
>
> The biggest advantage of the CUS3D dataset compared with other datasets is that it offers multiple types of data, including point clouds, meshes, and 2D images, with high-precision semantic labels provided by the official source. All the semantic annotations provided have undergone multiple rounds of manual cross-checking to eliminate any error, thereby avoiding semantic mistakes that researchers may encounter during data conversion. However, we did not focus on proposing a new annotation technique or tool.

---

> ### Author Response · Authors · 2023-11-22
> **Response to Reviewer ta5R (2/6)**
>
> Thanks for the valuable comments. We provide explanations below to help reviewers better understand our work.
>
> A2:
>
> Regarding 2D image semantic labeling, we adopted the ABAVA data engineering platform developed by an outsourced company in our project team. The data labeling module of this platform provides an automated instance segmentation tool based on the SAM algorithm. During the labeling process, the samples were first pre-labeled using the automated labeling module, and then manual adjustments were made to achieve high-precision pixel-level 2D image semantic labeling.
>
> We adjusted and merged the semantic annotation of 2D image data into 10 classes aligned with 3D labels to achieve consistency between 2D and 3D data. We also conducted baseline testing experiments on 2D images using the deeplabv3+ model. The experimental results will be supplemented in the paper in the near future.

---

> ### Author Response · Authors · 2023-11-22
> **Response to Reviewer ta5R (3/6)**
>
> Thanks for the valuable comments. We provide explanations below to help reviewers better understand our work.
>
> A3:
>
> The number and semantic categories of point-cloud datasets like Campus3D and SensatUrban are now large and diverse, respectively. However, the number of publicly available outdoor scene mesh semantic segmentation datasets is relatively small. As mentioned in Section 2, existing outdoor scene mesh semantic segmentation datasets such as ETHZ CVL RueMonge and Hessigheim have many labeling errors and unmarked regions owing to multi-view optimization and fuzzy boundaries. The SUM dataset only has six semantic categories, which is fewer compared with our CUS3D dataset. The contribution of the CUS3D dataset to the community is not to provide the point-cloud dataset with the most semantic labels, but as a comprehensive semantic segmentation dataset with multiple types of 3D data, such as point cloud and mesh. It provides the community of researchers with a benchmark dataset with various official data types, avoiding errors caused arising out of the sampling and data conversion processes. For example, the semantic labels of the mesh data in the Hessigheim3D dataset were transferred from the point-cloud data labels, resulting in approximately 40% of the regions not being labeled. The semantic labels of both point-cloud and mesh data in the CUS3D dataset underwent manual cross-checking, ensuring no labeling errors or unmarked regions, thereby providing high-precision datasets for the community of researchers to use.

---

> ### Author Response · Authors · 2023-11-22
> **Response to Reviewer ta5R (4/6)**
>
> Thanks for the valuable comments. We provide explanations below to help reviewers better understand our work.
>
> A4:
> The transformer architecture in semantic segmentation has shown strong performance, so it could be used in a new baseline experiment. We chose to add the stratified transformer as a baseline experiment to test the CUS3D dataset.  Please refer to Table 2 and Table 14 in the paper for the experimental results.

---

> > ### Author Response · Authors · 2023-11-22
> > **Stratified Transformer baseline experimental results**
> >
> > |     | Ground    | Building    | Road    | High vegetation    | Water    | Car    | Playground    | Farmland    | Grass    | Building site   |
> > | ---- | ---- | ---- | ---- | ---- | ---- | ---- | ---- | ---- | ---- | ---- |
> > | Acc    |91.22 | 94.43 | 96.12 | 94.67 | 99.34 | 99.21 | 99.85 | 97.89 | 96.12 | 97.88|
> > | Recall    | 33.44 | 90.1 | 83.3 | 91.56 | 59.32 | 57.01 | 14.55 | 62.45 | 55.78 | 18.32|
> > | F1 score    | 41.34 | 90.22 | 66.19 | 90.44 | 62.63 | 52.78 | 12.89 | 57.85 | 57.03 | 22.68|
> > | iou    | 60.23 | 70.02 | 75.44 | 83.36 | 58.47 | 58.66 | 32.45 | 39.66 | 34.59 | 39.33 |
> >
> > | mIou    | OA    | mAcc    |
> > | ---- | ---- | ---- |
> > | 55.22| 86.45| 96.67 |

---

> ### Author Response · Authors · 2023-11-22
> **Response to Reviewer ta5R (5/6)**
>
> Thanks for the valuable comments. We provide explanations below to help reviewers better understand our work.
>
> A5：
>
> Regarding 2D image semantic labeling, we adopted the ABAVA data engineering platform developed by an outsourced company in our project team. The data labeling module of this platform provides an automated instance segmentation tool based on the SAM algorithm. During the labeling process, the samples were first pre-labeled using the automated labeling module, and then manual adjustments were made to achieve high-precision pixel-level 2D image semantic labeling.

---

> ### Author Response · Authors · 2023-11-22
> **Response to Reviewer ta5R (6/6)**
>
> Thanks for the valuable comments. We provide explanations below to help reviewers better understand our work.
>
> A6：
>
> In all baseline experimental settings, we tried to maintain consistency in operations. Please refer to table 7 in the paper for the specific experimental parameters.
>
> For processing large-scale point-cloud data, the main tasks included data cleaning before annotation and data down-sampling during baseline experiments. Point-cloud data may contain noise, missing or outlier points. To improve the data quality before annotation, we performed outlier removal, smoothing, and filtering operations. Further, data down-sampling was done to achieve uniform point spacing. The original sample had a point spacing range of 0.4–0.6 m, so the downsampling parameter was set to 0.5 m. The goal was to achieve uniform point spacing without significantly altering the original number of point clouds.

---

> > ### Author Response · Authors · 2023-11-22
> > **Baseline experimental parameter settings result**
> >
> > |      |Epoch    | Batch_size    | Num_point    | Learning_rate    | Optimizer    | Momentum    | Parameters    |Time   |
> > | ---- | ---- | ---- | ---- | ---- | ---- | ---- | ---- | ---- |
> > | PointNet    |100| 24| 4096| 0.001| Adam| 0.9| 0.97M| 6.4h|
> > | PointNet++    |200| 32 | 4096 | 0.001 | Adam | 0.9| 1.17M | 6h |
> > | RandLA-Net    | 100 | 16 | 4096 | 0.01 | Adam | 0.9 | 4.99M | 7.5h |
> > | KPConv   |500 | 10 | - | 0.01| Adam | 0.9 | 14.08M | 8h|
> > | SPGraph   | 500 | 2 | 4096 | 0.01 | Adam | 0.9 | 0.21M | 5.5h |
> > | SQN    | 100 | 48 | 4096 | 0.01 | Adam | 0.9 | 3.45M | 7h |
> > | Stratified Transformer    | 500 | 8 | 4096 | 0.001 | Adam | 0.9 | 34.63M | 41h |

---

### Official Review · Reviewer_WLD1 · 2023-10-30

**Soundness:** 3 good
**Presentation:** 3 good
**Contribution:** 4 excellent
**Rating:** 6
**Confidence:** 4

**Summary:**

This paper introduces a new urban-scale 3D dataset. The dataset consists of both large-scale point clouds, meshes, and 2D images. A number of baseline methods have been evaluated on the dataset, and it shows that such a new dataset is still challenging for existing methods to learn 3D semantics.

**Strengths:**

1. Unlike most of existing datasets which only provide 3D point clouds, the introduced new dataset also provides 3D meshes for the community. In addition, it also has 2D images together with 2D annotations provided, which would be very useful for potential multimodal learning tasks.

2. The paper sets up the benchmark by evaluating 6 representative methods for 3D semantic learning, which looks great for future researchers.

**Weaknesses:**

The new urban-scale dataset looks great and would be beneficial for the community. Nevertheless, there are a number of minor questions:

1. There is a lack of details about the 2D images. For example, does every image has poses annotated? what is the image resolution and sampling density over the 3D arial space? How many 2D images in total? Does the dataset provide the exact correspondences between 2D pixels and 3D points/meshes?  I believe these would be critical if future uses want to fuse both RGB and 3D data for better semantic learning, or even 3D urban-scale novel view rendering.

2. There is a lack of details about the 3D reconstruction techniques. For example, how to find the pixel correspondences before triangulation? How to identify the outliers during 3D reconstruction? How to convert the 3D points to meshes? How about the quality of connected triangle meshes?

3. For 3D semantic annotation, the paper states that "according to the standard definition of semantic categories". What is the standard? In fact, it seems the categories "Road" and "Ground" are quite similar. Therefore, more specifications need to be added to justify your definition of classes. In addition, the paper states that "assign labels using annotation tools". What are the tools? and what are the annotation strategies?

4. For 2D semantic annotation, why are only 4336 images annotated? Is it a very small subset of the entire 2D image sequences? Besides, why are there 18 semantic classes on images, but 10 classes on 3D data? Are the 2D annotations aligned with 3D annotations? How about the quality of 2D semantic labels?

5. Figure 2 is a bit blurring.

6. Do the authors get all permissions to release the collected dataset including all 2D/3D data and annotations?

**Questions:**

Provided in Weaknesses.

---

> ### Author Response · Authors · 2023-11-22
> **Response to Reviewer WLD1 (1/5)**
>
> Thanks for the valuable comments. We provide explanations below to help reviewers better understand our work.
> A1:
> 1. We selected 4,336 images for detailed semantic annotation, including those taken with a 90° vertical-direction lens and one of the lenses tilted at a 45° angle.
> 2. The resolution of each image was 6144×4096 , with a point-cloud density of 53 points per m2 in 3D space and a mesh resolution of 3 cm.
> 3. We used the SHARE PSDK 102s camera, which is equipped with five lenses, one of which points vertically downwards at 90°. The other four lenses are tilted at a 45° angle in four different directions. Each lens was used to capture 2,168 images, resulting in a total of 10,840 2D aerial images.
> 4. The correspondence between 2D pixels and 3D points was obtained through affine transformation using intrinsic and extrinsic matrices. We can provide a 3D reconstruction report, pose parameters for each image, and camera intrinsic data in the subsequent data release. With this data, we can accurately establish the correspondence.

---

> ### Author Response · Authors · 2023-11-22
> **Response to Reviewer WLD1 (2/5)**
>
> Thanks for the valuable comments. We provide explanations below to help reviewers better understand our work.
> A2:
>  how to find the pixel correspondences before triangulation?
> To obtain the correspondence between 2D pixels and 3D points, we constructed an affine transformation. First, we computed the relative pose transformation between cameras by matching feature points in the images, and we constructed the extrinsic matrix to establish the relationship between the 3D vertex coordinate system and camera coordinate system. Then, using camera parameters such as focal length, we constructed the intrinsic matrix to establish the relationship between the camera coordinate system and pixel coordinate system. Through coordinate transformation, we established the relationship between 2D pixels and 3D points.
>
> How to identify the outliers during 3D reconstruction?
> During the feature matching stage, the matching error of feature points between two images was eliminated using the RANSAC algorithm to remove outliers, avoiding impact on the pose estimation process. During the dense matching stage, the point-cloud data were denoised and smoothed to remove outliers and sparse point clouds.
>
> How to convert the 3D points to meshes?
> We used the Delaunay triangulation algorithm to traverse the 3D point cloud and directly construct a triangular mesh from the point cloud.
>
> How about the quality of connected triangle meshes?
> The quality of the triangular mesh was determined by the quality of the point cloud the meshing method used. We used the Delaunay triangulation algorithm, which minimizes the radius of the circumscribed circle of each triangle. This algorithm tends to generate triangles that are as close to equilateral as possible, reducing the occurrence of elongated triangles and maximizing the minimum angle and average inscribed circle of all triangle faces. In the process of meshing the point cloud, we introduced a constrained Delaunay triangulation algorithm to solve the problem of meshing a point cloud with segment connection constraints, such as points lying on the same line or having parallel relationships. This improved the quality of the mesh.

---

> ### Author Response · Authors · 2023-11-22
> **Response to Reviewer WLD1 (3/5)**
>
> Thanks for the valuable comments. We provide explanations below to help reviewers better understand our work.
>
> A3:
>
> The CUS3D dataset abandons the previous fixed urban semantic labeling strategy and adopts a dynamic urban semantic labeling strategy that considers both fully developed and developing semantic categories. For example, the categories of “road” and “ground” may have some overlapping characteristics. However, “road” belongs to fully developed functional objects, which can be used for urban transportation planning and peak traffic control optimization research. Moreover, “ground” belongs to undeveloped objects, and labeling and recognition can help with early planning judgments in urban development. “Building sites” belong to the category of ongoing semantic objects and can be transformed into building sites in future semantic updates. The semantic labeling strategy of the CUS3D dataset considers the application, functionality, and temporal aspects of objects, making it more suitable for practical applications such as urban planning, transportation planning, and construction decision-making.
>
> The 3D semantic annotation tool used in this dataset is the labeling module in DP-modeler. The annotation strategy employed was multi-person collaborative quality control annotation. The entire dataset was labeled by three annotators with a quality control process in place. Each annotator simultaneously annotated one part of the dataset, and after completing their respective areas, they submitted them to the other two annotators for cross validation. After cross validation, a quality control expert conducted comprehensive verification to achieve high-quality semantic annotation results.

---

> ### Author Response · Authors · 2023-11-22
> **Response to Reviewer WLD1 (4/5)**
>
> Thanks for the valuable comments. We provide explanations below to help reviewers better understand our work.
>
> A4:
>
> The entire 2D image sequence consisted of 10,840 images from five different angles. Owing to the similar poses of the four cameras, having a tilt angle of 45°, we only selected the images from the sequence that included a 90° overhead view and a 45° tilted view for semantic annotation. The annotated image sequence accounted for 40% of the entire image collection.
>
> After collecting the aerial image sequence initially, we expected to use these finely annotated 2D raw images for 3D reconstruction to obtain 3D data with semantic meaning. However, the high similarity of pixel features after semantic labeling would make it difficult to match feature points in the triangulation process. Therefore, we decided to follow the traditional reconstruction method, first reconstructing the original 3D data, and then performing semantic annotation. We also merged the initial 18 semantic categories into 10 categories that were easier to understand.
>
> Considering the suggestions from the expert reviewers, we further aligned the semantic categories of 2D images with the 10 semantic categories of 3D data, ensuring the alignment of annotations between 2D and 3D. Please refer to the paper for the results of 2D image semantic modifications.
>
> The semantic labels of 2D images had high accuracy and utilized pixel-level annotation.

---

> ### Author Response · Authors · 2023-11-22
> **Response to Reviewer WLD1 (5/5)**
>
> Thanks for the valuable comments. We provide explanations below to help reviewers better understand our work.
>
> A5&6:
>
> We made Figure 2 clearer. We have made the revisions in the paper. Our team has full permission to access all the data in this dataset and can openly publish it.

---

### Author Response · Authors · 2023-11-22
**Official Comment by Authors**

We greatly appreciate the professional feedback from all the reviewers.

We have carefully considered your valuable suggestions and made necessary revisions. We have addressed each question from every reviewer and the revised content is highlighted in blue text in the latest version of the paper. If you have any further questions or feedback, please feel free to contact us. Thank you!

Note: The experiments regarding the benchmark testing of 2D image data are still in progress, and the experimental results will be supplemented and finalized in the paper before the submission deadline.